# Studies of Monoamine Neurotransmitters at Nanomolar Levels Using Carbon Material Electrodes: A Review

**DOI:** 10.3390/ma15165782

**Published:** 2022-08-22

**Authors:** Pankaj Kumar, Isha Soni, Gururaj Kudur Jayaprakash, Roberto Flores-Moreno

**Affiliations:** 1Laboratory of Quantum Electrochemistry, School of Advanced Chemical Sciences, Shoolini University, Bajhol, Solan 173229, India; 2Department of Chemistry, Nitte Meenakshi Institute of Technology, Bangalore 560064, India; 3Departamento de Química, Universidad de Guadalajara, Blvd. Marcelino García Barragán 1421, Col. Olímpica, Guadalajara 44430, Mexico

**Keywords:** carbon materials, neurotransmitters, voltammetric techniques, oxidation–reduction, surface-modified electrodes, electrochemical detection

## Abstract

Neurotransmitters (NTs) with hydroxyl groups can now be identified electrochemically, utilizing a variety of electrodes and voltammetric techniques. In particular, in monoamine, the position of the hydroxyl groups might alter the sensing properties of a certain neurotransmitter. Numerous research studies using electrodes modified on their surfaces to better detect specific neurotransmitters when other interfering factors are present are reviewed to improve the precision of these measures. An investigation of the monoamine neurotransmitters at nanoscale using electrochemical methods is the primary goal of this review article. It will be used to determine which sort of electrode is ideal for this purpose. The use of carbon materials, such as graphite carbon fiber, carbon fiber micro-electrodes, glassy carbon, and 3D printed electrodes are only some of the electrodes with surface modifications that can be utilized for this purpose. Electrochemical methods for real-time detection and quantification of monoamine neurotransmitters in real samples at the nanomolar level are summarized in this paper.

## 1. Introduction

The chemical messengers called neurotransmitters influence a large number of physiological and psychological human bodily functions. Because they are directly related to the central nervous system (CNS), they directly handle neurophysiological exercises, such as emotions, sleep, memory, and other cognitive functions by amplifying, transferring, and converting signals through a synapse to a target cell. Defective NTs in the CNS cause a slew of diseases, including Huntington’s syndrome, Parkinson’s, Alzheimer’s disease, autism, and epilepsy. Additionally, it can cause psychotic diseases, such as dementia, attention deficit hyperactivity disorder, schizophrenia, and depression, as well as dejection, anguish, congestive heart failure, glaucoma, arrhythmias, and sudden infant death syndrome. In 1921, German pharmacologist Otto Loewi identified the first extensively detectable neurotransmitter, acetylcholine (Ach). More than 500 NTs were found from 1921, but the actual number of these distinct NTs in humans has yet to be determined. NTs are stored in synaptic vesicles and released into the synaptic cleft and diffused across it, where they get attached to specific receptors on the post-synaptic neuron’s membrane. This bonding perhaps exerts excitation (depolarization) or inhibitory (repolarization) control over the post-synaptic neuron. Immunocytochemical approaches that identify the site of the transmitter material are commonly used to determine the anatomical localization of NTs. Many transmitters, primarily neuropeptides, are co-localized, indicating that a neuron can release multiple transmitters from its synaptic terminal, according to immunocytochemical techniques. Neurotransmitters can be classified in a variety of ways. It is sufficient to divide them into monoamines, amino acids and peptides for certain classification purposes [1,2]. We mainly focus on hydroxyl group-based monoamine neurotransmitters as mentioned in Table 1.

### 1.1. Dopamine

Dopamine (DA, chemically 3,4-dihydroxyphenethylamine) is a hormone and neurotransmitter that operates in the hormonal, central, renal, and cardiovascular systems. It belongs to the catecholamine and phenethylamine families of chemical molecules. Abnormal DA concentrations in humans are linked to neurodegenerative illnesses, such as Alzheimer’s, Parkinson’s, and schizophrenia. Around 80% of the catecholamine content in the brain is dopamine, which is an amine created by removing the carboxyl group from a molecule of L-3,4-dihydroxyphenylalanine (L-DOPA), a neurotransmitter produced in the brain and kidneys. The effects of this biomolecule’s abnormal concentration also highlights its role in proper brain functioning. In recent years, a huge number of studies were published on DA quantification employing electrochemical sensors with acceptable selectivity, detection limit, and sensitivity [3].

### 1.2. Serotonin

Serotonin (ST), also known as 5-hydroxytryptamine, or 5-HT, is a monoamine-type neurotransmitter that is produced in the brain, gut, and spinal cord. It regulates a wide range of behavioral and physiological activities, including mood, sleep, emesis, addiction, liver regeneration, infantile autism, appetite, temperature, eating behavior, sexual behavior, movements, and gastrointestinal mobility, with 14 variants of serotonin receptors. Depression, anxiety, migraines, uncontrolled hemostasis, blood clotting, sudden infant death syndrome (SIDS), and carcinoid syndrome are all associated with low levels of ST [4]. With the plethora of diseases that an aberrant amount of 5H-T in extracellular fluids might cause, it is clear that substantial research on its detection in clinically oriented research is warranted. Other approaches (chromatography, mass spectrometry, and electrophoresis) were tried in the quantification of ST, just as they were in the measurement of every other neurotransmitter, but they are ineffective. In the following sections, the design of the electrode, selectivity, and limit of detection (LOD) of several electrodes constructed for the measurement of ST will be thoroughly examined [5,6].

### 1.3. Epinephrine

Epinephrine (EP) is a hormone and medication which is mostly involved in visceral functions, such as respiration, and is also known as adrenaline. Chemically, it is 4-[(1R)-hydroxy-2-(methylamino)ethyl]benzene-1,2-diol, belonging to the catecholamine class of NTs. Takamine was the first to isolate EP from the adrenal hormone in 1901, while Friedrich Stolz and Henry Dakin were the first to synthesize it in the laboratory. It allows the body to prepare for flight or battle by boosting blood flow to the muscles. Epinephrine is a neurotransmitter produced by the adrenal glands and is required by the neurological system. Additionally, while the existence of adrenaline is necessary for human survival, it is released from the body when a person is stressed. The effects of an adrenaline overdose on the human body include sudden numbness, slurred speech, and exacerbated breathing problems. Because of its control over the neurological system’s performance, an upset in the balance of EP in the human body fluid to a level outside of 0.09–0.69 mg mL^−1^ in a control range leads to the exposure and emergence of various chronic disorders (such as Parkinson’s disease). Many articles highlighted developments of electrode sensors for the detection of epinephrine over several years. The detection limit, electrode manufacturing, stability, and electrode selectivity, which are discussed in the following sections, highlight the advances accomplished thus far [7].

### 1.4. Norepinephrine

Norepinephrine (NE), a catecholamine NT found in the CNS of mammals and considered the most important one that regulates the circulatory system, relieves pain, detects stress, and depression, and regulates appetite. Chemically, it is 4-[(1R)-2-amino-1-hydroxyethyl]benzene-1,2-diol, and changes in NE plasma levels caused by metabolic dysfunction can result in a variety of clinical diseases. NE works on target cells by activating noradrenergic receptors on cell surfaces, resulting in increased heart rate, anxiety, bladder inhibition, and gastrointestinal motility. Hypoglycemia, anxiety, high blood pressure, paleness, and headaches are all symptoms of a disturbed NE balance in the body. As a result, measuring the amount of NE in the extracellular fluid (ECF) can be used to diagnose a variety of diseases. For the first time, modified carbon electrodes with high selectivity, stability, and sensitivity were effectively used to analyze injection samples of NE [7].

Because of their difficult immobilization method and the instability induced by the degradation of enzymatic activity over time, enzyme-based biosensors have disadvantages. Nonenzymatic sensors, on the other hand, can easily detect electroactive NTs. Even though these NTs bio-sensors are low-cost, stable, and biocompatible for most uses, their selectivity and specificity are still under question. A large number of the biochemical genus has equivalent redox potentials in the brain, making it challenging to use nonenzymatic NTs bio-sensors, which are sensitive to these types of species [8]. For this, a variety of ways are documented. There are a variety of methods reported for the detection of neurotransmitters, including HPLC, GC–MS (gas chromatography–mass spectroscopy), and electrochemical sensors. Electrochemical approaches have gotten a lot of interest due to their advantages of fast detection, low cost, portability, ease of operation, excellent selectivity, and excellent efficiency, among other things [9]. Chemically modified electrodes (CMEs) have lately piqued interest in the electrocatalytic field. Essential redox systems, oxidation/reduction, and functioning mechanisms of such electrodes are determined by many factors. On the qualities of the modified materials that were utilized to increase target selectivity and sensitivity. This electrode is inexpensive and has many benefits, including broad potential windows (both anodic and cathodic region), a small background current, ease of fabrication, and rapid surface renewal. The most essential aspects of CMEs are their potential to speed up the electrochemical process by lowering the overpotential and improving electron transfer kinetics when compared to an unmodified electrode. The use of these modified electrodes is very demanding due to the features, such as less noise, a low residual current, broad redox potential ranges, ease of production, low cost, and renewability [10]. Electrodes modified using different chemicals were employed for the detection of NTs in many research publications published in the literature. In this review, we aim to compile the necessary hydroxyl group-based monoamine neurotransmitters and their detection using different electrodes with modifications.

Compiling these reports into a review paper is difficult because most research studies do not provide a critical comparison with previously published work. From the beginning to the present, we thoroughly presented the most valuable, promising, accurate, and successful electrochemical techniques for the sensing of these monoamines. According to us, there is no review study published that summarizes the electrochemical methods for monoamine neurotransmitters’ determination.

This review article gathered many reviews and research articles for NTs’ detection with surface-modified electrodes (SME) based on polyelectrolytes and conducting polymers, metal and carbon nanoparticles, ionic liquids, and nanocomposites, as compared to the analytical results of different techniques. Among the best NTs’ detection approaches are linear working concentration range, selectivity, sensitivity, the lower detection limits (LODs), sample prearrangement simplicity, stability of the electrode surface that has been changed, and signal intensification over bare electrodes, as well as the separation between oxidation peak potentials [11]. The interference used includes uric acid (UA), ascorbic acid (AA), 5-hydroxy indole acetic acid (HIAA), L-DOPA, tryptophan, tyrosine, alanine, glutamic acid, histamine, aspartic acid, oxalic acid (OA), glucose, cysteine, 3,4-Dihydroxyphenylacetic acid (DOPAC), etc.

## 2. Comparison of Structures and Mechanism of Monoamine NTs Based on the Position of the Hydroxyl Group

All four monoamines have the presence of hydroxyl groups (Figure 1) which are responsible for the electron oxidation in the presence of another hydroxyl radical (OH*). There may be a transfer of one or two electrons to obtain the oxidized product. In the case of dopamine, it may present in either a protonated or deprotonated form. The reaction of protonated dopamine with OH* is carried out by adding to an aromatic ring, while deprotonated dopamine proceeds via direct one-electron oxidation to form a o-semiquinone radical or o-semiquinone. The above mechanism is quite similar to all other monoamines (Figure 2) while the rate of reaction may vary. The position of the hydroxyl groups in the benzene ring relative to the amine group is responsible for varying the rate of reaction and ultimately the sensing of these monoamine NTs using various electrodes changes considerably.

## 3. Neurotransmitter Biosensing

On the target cell, a neurotransmitter can have an excitatory, inhibitory, or modulatory action. The receptors that the neurotransmitter interacts with at the post-synaptic membrane determine the effect (Figure 3). A neurotransmitter regulates trans-membrane ion flow to either raise (excitatory) or decreases (inhibitory) the likelihood of an action potential being produced by the cell with which it comes into contact [12]. Type I synapses contain receptors that have excitatory effects, whereas Type II synapses contain receptors that have inhibitory effects. Despite the enormous variety of synapses, they all send only these two types of information. The two types have distinct appearances and are primarily found in various sections of the neurons that are affected by them. All synaptic membranes contain modulatory receptors, and the binding of neurotransmitters initiates signaling cascades that assist the cell to regulate its activity. The binding of neurotransmitters to receptors that have modulatory effects can result in a variety of outcomes. By recruiting more or fewer receptors to the synaptic membrane, it may result in an increase or decrease in sensitivity to the subsequent stimuli [13].

Real-time detection of NTs is of tremendous interest in neuroscience because they perform key roles in the central nervous system and influence the severity of neurodegenerative disorders [14]. Billions of neurons communicate through trillions of synapses and electrical channels in the human brain. When exocytotically released NTs establish a concentration gradient between neurons, the NTs function as ligands that attach to receptors on neighboring cells in a reversible manner, creating a conformational change that causes a sequence of biochemical responses within the cell. NT detection takes place in the extracellular space due to the nanometer scale of the synapse and the micron-scale of electrode probes. Because of the quick release and clearance of NT, low concentration levels, and the presence of interfering analytes in the extracellular space, most standard analytical methods continue to struggle to quantify it [15]. The various modified electrodes are used to sense these neurotransmitters electrochemically.

## 4. Neurosensing via Electrochemical Studies

Electrochemical studies or electrochemistry is the study of various techniques to analyze the chemical reactivity using electrical simulations by detecting mainly the oxidation and reduction mechanism in a particular reaction [16,17]. Potential, charge, current, and time are the four main experimental key frameworks in an electrochemical experiment that are typically measured. There are so many amalgamations of working electrode types, and these parameters are achievable in electrochemistry, while several techniques are possible using principles of electrochemistry. We consider mainly cyclic and differential pulse voltammetry to detect the monoamine neurotransmitters. The cyclic voltammetry technique is used to investigate chemical pathways involving electron transfer [18]. Because this method includes linearly altering an electrode potential between two limits at a certain pace while monitoring the current that develops in an electrochemical cell, it is important and advantageous. It provides a very useful tool for understanding and studying the redox-type chemistry of molecules applicable to biological redox reactions too. While in the case of differential pulse voltammetry, it is more sensitive than CV and other techniques because it has a low capacitive current which ultimately leads to very high sensitivity. Because of the tiny step sizes in DPV, voltammetric peaks are narrower, and DPV is frequently employed to distinguish analytes with comparable oxidation potentials [19,20].

### 4.1. Cyclic Voltammetry (CV)

CV is, no doubt, a perfectly adaptable and versatile electroanalytical technique used for the examination of electronically active species. The fundamentals of CV consist of cycling the potential of an electrode that is immersed in a stable unstirred solution and used to measure the final resulting current. There is a setup of the three-electrode system which mainly includes working, reference, and counter electrodes. The potential of the working electrode is controlled against reference electrodes e.g., saturated calomel electrode (SCE) or Ag/AgCl electrode with Pt-wire as the counter electrode. The potential, called controlling potential, is applied across all electrodes to generate a signal considered an excitation signal, which is a linear potential scan with a triangular waveform for CV. It primarily requires a waveform generator to generate the excitation signal and a potentiostat to apply it to an electrochemical cell, as well as a current-to-voltage converter to measure the generated current, and an XY recorder or oscilloscope to display the voltammogram (Figure 4) [21].

### 4.2. Differential Pulse Voltammetry (DPV)

The principle of the DPV approach is to measure the difference in the rate of decay of charging and faradaic currents when the potential pulse is applied. The faradaic current is applied as a function of time during scanning. The potential is scanned with a series of pulses, while the current is monitored at the beginning and end of each pulse, which is fixed at a small amplitude while the current generates peak-shaped voltammograms (Figure 5). We can also use this technique to estimate the qualitative and quantitative value of redox reactions and to characterize the performance or structure of microbial bio-film formed on electrodes because of its high sensitivity. The setup of the electrode system is the same as used in CV studies [22].

## 5. Sensing of Monoamine NTs Using Surface-Modified Electrodes

### 5.1. Dopamine

#### 5.1.1. Carbon Paste Electrode (CPE) Modified with Activated Carbon

To make a functional electrode, graphite powder and activated carbon were combined. The finally prepared paste was then pressed into the cavity of the electrode and polished with smooth paper. A carbon bar was used to make electrical contact. This prepared electrode is named here AC-CPE. Electrochemical studies using cyclic and differential pulse voltammograms were run out in a traditional electrochemical cell of the three-electrode system using a DAQecorder/potentiostat EA163 controlled by eDAQEChem data gathering software. The counter, reference, and working electrodes were Pt-wire, solution of Ag/AgCl/KCl, and activated carbon modified carbon paste electrode (AC–CPE), respectively. To begin with, the electrochemical behavior of the changed electrode revealed that the majority of the cathodic and anodic peak potentials were positively and negatively displaced, respectively, with an increase in the DA current peak. The diffusion coefficient (D = 1.01 × 10^−5^ cm^2^ s^−1^) and kinetic parameters for dopamine at the surface of the AC-CPE, such as the electron transfer coefficient (a = 0.48), apparent electron transfer rate constant (ks = 0.80 s^−1^), and catalytic reaction rate constant (k_cat_ = 40.90 104 mol L^−1^ s^−1^), were then determined using electrochemical methods. The AC-CPE exhibited favorable electron transfer kinetics and electro-catalytic activity when it came to dopamine oxidation/reduction. The CV and DPV experiments were studied in the potential ranges of 0.5 and 1.0 V and 0.3 to + 0.7 V, respectively. Voltammograms were recorded using carbon paste electrodes that were both unmodified and AC modified. There were no oxidation and reduction peaks of the CV curve on an anodic and cathodic branch when the absence of DA at CPE and AC-CPE. Meanwhile, enhancement by one and a half times in the peak current when connected to a modified electrode shows the result that AC carbon has high electro-catalytic activity towards oxidation–reduction in dopamine (DA). With a detection limit of 3.09 × 10^−8^ mol/L, DPV was effectively employed to determine DA in the linear response range of 1.0 × 10^−7^ to 1.0 × 10^−3^ mol/L [23].

#### 5.1.2. CPE Modified with Polyalizarin Yellow R (PAYR)

The prepared CPE surface, a mixture of graphite powder, and oil, at a ratio of 5:0.7 (*w*/*w*), was polished with a simple filter or weighing paper, and then rinsed twice with double-distilled water. Then, using cyclic sweeping, PAYR was deposited electrochemically on the modified surface of the CPE. The surface-modified electrode was washed thoroughly with double-distilled water after each measurement, and then repeatedly scanned in a buffer solution of phosphate at pH 7.0 having a potential range from −0.4 V to 0.8 V. An electrochemical workstation CHI 660D was used to conduct all the experiments. (Chenhua Instruments, Shanghai, China). A bare and modified carbon paste electrode was used as a working electrode, whereas the counter electrode and reference electrode were a platinum wire and a saturated calomel electrode (SCE), respectively. The CV study of dopamine is conducted by using a PAYR/CPE electrode. The cyclic voltammetric responses in equimolar 5.0 mM solution of K_3_Fe(CN)_6_ and K_4_Fe(CN)_6_ at the bare CPE surface and PAYR/CPE were measured with the supporting electrolyte made up of 0.1 M KCl. When the film of PAYR is changed on the bare CPE surface, there is a great increase in the peak current and a dramatic drop in the peak-peak separation between anodic and cathodic peaks. The CV results show that the PAYR film is conductive, the linear calibration curve was obtained as 0.449–70.1 mM, and the detection limit of 0.16 mM with a wide linear dynamic range, high sensitivity, and selectivity [24].

#### 5.1.3. CPE Modified with Sulfated β-Cyclodextrin (S-CD)

A Solartron potentiostat was used to conduct cyclic voltammetry (CV) experiments (model SI 1285A). A Chi440 Shanghai, China potentiostat was used to perform differential pulse voltammetry (DPV) (Model 400). Macro electrode investigations included the system of three electrodes comprised primarily of an unmodified CPE or S-CDCPE as the working electrode, a saturated calomel as a reference electrode, and a platinum wire as the counter electrode. For the micro electrode investigations, the working electrode was an unaltered CPE or S-CDCPE, while a silver–silver chloride and silver wire served as the reference and counter electrodes, respectively. To make the modified carbon paste electrode, an agate mortar was used to fully hand-mix the necessary amount of S-CD and silicon oil with 0.71 g graphite powder for around 30 min, yielding a homogenous S-β-CD modified carbon paste. The electrode performs well in DA electrochemical oxidation, with a detection range of 5 × 10^−7^ M to 5 × 10^−4^ M and a detection limit of 1.33 × 10^−7^ M. It was revealed that the developed sensor’s sensitivity for detecting DA is proportional to the amount of S-β-CD in the paste. The data show that there is a 10-fold increase in the oxidation current form S-β-CDCPE responsiveness for DA as compared to the bare unmodified CPE [25].

#### 5.1.4. Pencil Graphite Electrode (PGE) Modified with Uric Acid and Ascorbic Acid at Poly(Xylenol Orange)

In an electrochemical cell, 5 × 10^−4^ M Xylenol Orange (XO) was mixed with a supporting electrolyte made up of 0.2 M buffer solution of phosphate at 10 pH to make poly(XO)/PGE. At a scan rate of 100 mV/s with 10 repeated cycles, the voltage was maintained between 400 mV and 2000 mV. On the surface of PGE, a homogenous poly(XO) layer may be produced. The electrode was then cleaned with double-distilled water before being maintained in 0.2 M PBS at pH 7. A model-201 electroanalyser was used for the electrochemical studies (EA-201 chemilink system). In the electrode setup, a bare graphite pencil electrode with 0.5 mm in diameter was the working electrode, a platinum wire was the counter electrode, and saturated calomel was the reference electrode. The redox peaks produced with the modified electrode had a significant improvement. A diffusion-controlled electrode mechanism was discovered to be responsible for the scan rate effect. The electrochemical oxidation of dopamine was discovered to be pH-dependent, with a LOD value of 9.1 × 10^−8^ M. Using both CV and DPV techniques, the simultaneous investigation produced a good result with a significant potential difference between dopamine and other bioactive organic compounds [26].

#### 5.1.5. PGE Modified with Cu/Cu_x_O Nanoparticles

All electrochemical measurements and electrodeposition activities were performed using an Ivium/vertex potentiostat/galvanostat device coupled to a three-electrode cell. The reference and counter electrodes were an Ag/AgCl (3M KCl) electrode and a platinum wire, respectively. Two cathodic peaks emerged, corresponding to the reduction in Cu^2+^ → Cu^1+^ and Cu^1+^ → Cu^0^, and exhibiting CuNP deposition on the surface of PGE. The peaks showing oxidation are also visible in the forward cycle, indicating that the reaction is reversible. The mechanism showing the oxidation of CuNPs in the basic solutions, such as NaOH, can be illustrated by the given reactions:Cu + OH^−^ → CuOH + e^−^
2CuOH → Cu_2_O + H_2_O
Cu_2_O + 2OH^−^ → 2CuO + H_2_O + 2e^−^

In 0.1 M NaOH solution, after 20 voltammetric cycles, CuNPs was converted to copper oxide. Using CV and DPV, the ability of modified electrodes to detect DA in 0.1 M phosphate buffer solution at pH 5.8 was evaluated. For DA electro-oxidation, the electrode was changed at −0.6 V vs. Ag/AgCl and 150 s and produced the maximum peak current (Ip). In comparison to biosensors modified with rod-shaped CuO, ZnO, TiO_2_, or AuNPs, this enhanced electrode had a LOD value of 1.06 M and a superior sensitivity of 0.51 A/M.The best biosensor was easy to use and had good selectivity (E_UA_–E_DA_ = 0.14) [27].

#### 5.1.6. CPE Modified with Poly(Reactive Blue) and Lysine

The electrochemical investigation of dopamine is conducted in the presence of uric acid and ascorbic acid. Along with ascorbic and uric acid, for the preparation of electrodes, graphite powder, silicon oil, and reactive blue were used. The electropolymerization technique was used to fabricate poly(reactive blue) MCPE i.e., the modified carbon paste electrode by using 1 mM aqueous reactive blue monomer, containing sodium hydroxide (0.1 M) as the supporting electrolyte. The voltammetric instrument CHI-660c model was used to perform the electrochemical experiments. A three-electrode single component cell system was made using bare CPE and MCPE as working electrodes, while using saturated calomel and platinum wire as reference and auxiliary electrodes, respectively. With a sweep rate of 100 mV/s, the electrochemical response of 10 μM DA at the BCPE and poly (Reactive blue) MCPE was investigated in pH 7.4 PBS (0.2 M) as a supporting electrolyte. The cyclic voltammogram of dopamine in bare CPE revealed lower current and greater redox peak potential differences (Ep) of 52 mV. Simultaneously, poly (Reactive blue) MCPE exhibited a significant increase in the redox peak current, with a 19 mV decrease in Ep. This lowers the overpotential and increases peak currents, revealing poly (Reactive blue) MCPE’s electrocatalytic activity in the electrochemical oxidation of dopamine.

The DPV technique was used for the interference study of the mixture of the sample with different concentrations containing dopamine, ascorbic acid, and uric acid. The concentration of one species was varying while the other remained constant, and vice versa. Dopamine concentration rose from 5 to 30 mM, but ascorbic acid (1 mM) and uric acid (5 mM) concentrations stayed constant. The voltammogram indicates a linear increase in anodic peak current with no shift in dopamine oxidation peak potentials, and no change in uric and ascorbic acid voltammetric responses (Figure 6 and Figure 7). This result demonstrates increased current sensitivity and the absence of the background current, which aids in the exact and accurate detection of dopamine at poly (Reactive blue) MCPE having a very low detection limit i.e., 4600 nM [28]. Gururaj et al. used lysine to modify the carbon paste electrode to enhance the sensitivity of the electrode for dopamine detection. The cyclic voltammetry technique is used to study the electrochemical behavior of dopamine on the LMCPE surface. The electron transfer (ET) processes between DA and the BCPE surface are slow, as demonstrated by the background capacitance current of DA at BCPE of 0.110 V. The capacitance current of DA at LMCPE, on the other hand, is 0.53 V, indicating that the ET reactions between DA and LMCPE are quick. As a result, lysine works as an electrocatalyst for DA redox reactions, and LMCPE is a more sensitive voltammetric sensor for detecting DA [29]. In another work, CTAB was used for the modification of a carbon paste electrode. The limit of detection and linear range observed for dopamine are 200 nM and 1000–70,000 nM, respectively [30].

Various electrodes employed for the electrochemical detection of DA are compiled in the Table 2 below.

### 5.2. Serotonin

#### 5.2.1. GCE Modified with Conducting Polyelectrolytes and Polymers

These active polymers have functional groups that can show redox reactions, allowing them to electrocatalyze the oxidation or reduction in the target molecule. Finally, with the attractive forces of the specified functional groups connected to the polymer, the analyte accumulates on the electrode surface. An ion exchanger is a polyelectrolyte that exits as a polycation or polyanion and, through the process of ion exchange, the ionic species get attracted to the surface from the bulk of the solution. The electrical properties of polyelectrolytes and conductive polymers are similar to those of polymers and metals. The specified functional groups on the surface of the polymer have a selective affinity for the analyte of interest via ion exchange capabilities, electrostatic interactions, or hydrogen bonding. These conductive materials prevent the electrode from fouling by forming a protective surface. The different approaches of GCE modified with various conducting polymers are given in Table 3 SWV, DPV, CV, and SW-ASV techniques were used to detect the serotonin with LOD ranges from 0.013–1700 nM using different materials [4].

#### 5.2.2. GCE Modified with Carbon Nanomaterials

Carbon nanomaterials outperform traditional electrochemical sensors in terms of electrocatalytic activity, electron transfer, biocompatibility, adsorption, and kinetics characteristics. Nanoparticles (NPs), such as carbon nanotubes and nanocomposites, as well as graphene, and other carbon nanomaterials, are utilized to modify the surface of bare electrodes to detect ST. Even when CNTs are in direct contact with water, their essential electrical characteristics are preserved [2]. CNTs were discovered to be useful as a surface coating material for electrochemical investigation as a result of this discovery [48,49]. To take advantage of the synergistic effect, MWCNTs and SWCNTs are widely employed, and they are often combined with other conducting materials. For the first time, Wu et al. employed carbon nanotubes to modify GCE and detect serotonin and dopamine simultaneously. With decreased oxidation overpotentials, the current responsiveness of DA and ST improved significantly. The multiwalled carbon nanotube/dihydropyran (MWCNT-DHP) modified GCE showed two well-defined sensitive oxidation peaks, whereas the bare GCE gave a merging peak for DA and ST [3]. To take advantage of synergistic effects, for surface modification, MWCNTs were coupled with ionic liquids and other good conducting materials. Sun et al. created a composite modified GCE electrochemical sensor based on ionic liquid [OMIM]PF_6_ and CNTs [50]. Satyanarayana et al. also created a sensor using MWCNTs and chitosan (CHT) with carboxylic group functionalized carboxylic groups. The serotonin molecules, which are positively charged, get attracted by functionalized MCWCNTs i.e., f-MWCNTs, with a negative charge, causing the increase in the diffusion rate. Due to the specific binding nature of the chitosan biopolymer, the film of nanocomposites looked much more stable for several studies conducted over a large period [51]. Ran et al. [52] created a GCE modified with a nanocomposite of poly(p-aminobenzene sulfonic acid), MWCNTs, and CHT using chitosan with functionalized MWCNTs. On the electrode’s surface, negatively charged poly(p-ABSA) and chitosan with a positive charge, combined with carbon nanotubes, create a well-conducting and more stable combination. When MWCNTs were employed with CHT, however, the detection was found to be inferior to that of MWCNT/DHP or MWCNT/IL gels. MWCNTs were also employed with nafion, a cation exchanger for catalytic oxidation of serotonin and dopamine with immobilization matrix capability [53]. As a result of the synergistic electrocatalysis effect, MWCNTs with Ni(OH)_2_ nanoparticles and nafion improved electron transport for the oxidation of serotonin [53]. Nafion also has a high resistance to mechanical damage. The modified surface had a significantly reduced detection limit. Because of selective electrocatalyzed oxidation of NE, benzofuran-derivative functionalized MWCNTs detect NE and ST concurrently [32]. Peak separation between NE and ST improves with a negative shift in NE’s peak potential. The sensor is not very useful for ST because the detection limit is not high enough (2000 nM). Graphene, because of its huge electrical conductivity, surface area, and high tensile strength, is a different type of carbon nanomaterial that is employed for the modification of electrodes. Kim et al. electrocatalyzed the oxidation of serotonin by reducing graphene nanosheets in various chemical circumstances and coating them on GCE. The conductivity of graphene sheets with a high oxygen content is higher, and the electric resistance is lower than those with low oxygen content. The best results yielded by the reduction of a sheet with ammonia and hydrazine solution had the lowest LOD value, highest sensitivity, broadest linear range, shortest response time, and best-characterized oxidation peak of ST [54]. Later, to increase the electrical conductivity of a serotonin sensor, graphene oxide (GO) was functionalized with polylactic acid and platinum [55]. Again in 2014, covalently functionalized the GO with an amino phenyl porphyrin derivative (GO-P). More active sites and a higher peak current value for serotonin were found in the electrochemically reduced GO-P (ERGO-P), as well as the ability to detect ST and DA concurrently in the presence of ascorbic acid. Porphyrin derivative functionalized GO produced the best results among graphene-based SMEs, with the largest linear range and lowest detection limit [4]. In Table 4 a comparison of different research methodologies based on the analytical figure of merits is offered.

#### 5.2.3. Metal or Metal Oxides Nanoparticles Modified GCE

Metal nanoparticles have properties different than bulk materials, such as unique optical, electronic, magnetic, and catalytic properties due to their small size and large surface-to-volume ratio with bioabsorbable capabilities. Lin and Li built a gold nanocluster electrodeposition electrochemical sensor on a polypyrrole (PPyox) screen modified GCE. The peaks of oxidation for ST, AA, and DA were successfully resolved into three sharp DPV peaks. The negatively charged groups on PPyox film are responsible for the serotonin accumulation on film: these results are due to preferential interconnection with coupled serotonin via pi–pi interaction [57]. In another study by Wei et al., the serotonin molecule is catalytically oxidized at an Au nanocluster. GNPs were permitted to self-assemble the GCE modified with L-cysteine and employed for serotonin sensing [58]. In the presence of DA, EP, AA, and FA, this unique SME allows for the selective measurement of ST. Engineered materials containing selective binding sites for target molecules are known as molecularly imprinted polymers (MIPs). Due to the deficiency of electrocatalytic activity and conductivity of MIP, these sensors have lower sensitivity. Xue et al. inserted gold nanoparticles in MIP to enhance conductivity. His team created a modified electrode sensor using GCE coated with a reduced graphene oxide/polyaniline (RGO/PA) nanocomposites double-layered membrane and MIPs implanted with gold nanoparticles. The incorporation of Au nanoparticles into MIP resulted in increased conductivity and electrocatalytic activity with a remarkably high sensitivity of 11.7 nM. Later, GCE was tweaked using a variety of metal and metal oxide-NPs [59].

Because of their high surface area and biocompatibility, TiO_2_ nanoparticles were employed as film-forming materials. Mazloum-Ardakani and Khoshroo described the production and implementation of functionalized MWCNTs with TiO_2_ NPs and 1-butyl-3-methylimidazoliumtetrafluoroborate (IL) as a very precise and sensitive sensor for simultaneous detection of serotonin and isoproterenol. As a result of synergism, combining CNTs with TiO_2_ improves the electrolytic and catalytic effects of both TiO_2_ and CNTs. However, as compared to Au NPs, the detection limit is lower [32]. Serotonin detection was conducted with Ag nanoparticles by Sadanandhan et al. The best working electrode with surface modifications is a PEDOT-reduced graphene oxide–silver hybrid nanocomposite, as compared to GCE biosensors based on all the metal or metal oxide nanoparticles for the detection of ST, with a much lower LOD i.e., 0.1 nM, the widespread linear range high operational stability, and reproducibility. Between RGO sheets, PEDOT (Poly(3,4-ethylene dioxythiophene)) nanotubes act as a conductive bridge, allowing electrons to flow more freely [60]. Irradiated metal oxide nanoparticles were employed to modify GCE by Anithaaa et al. The application of a 100 kGy radiation dosage to WO_3_ NPs generates remarkable changes in the crystal structure, external appearance, and peculiar surface area, as well as an increase in electron transfer rate. The constructed sensor GI-WO_3_/GCE was shown to have good electrocatalytic capabilities, great selectivity in the presence of synchronizing electroactive species, extended immense stability, exceptional and excellent repeatability, and an impressive limit of detection of 1.42 nM [61]. Dinesh et al. employed cobalt oxide nanoparticles in combination with RGO to alter GCE in one of their studies. A simple in situ electrochemical technique was used to create the modified electrode.

Metal ions with a positive charge are attracted to GOs with oxygen functions and become attached to the surface of the GO. The oxidation peak current of serotonin at Co_3_O_4_/GCE was reported to be 4.1 times greater than that found at RGO/GCE, and 2.4 times higher than that found at Co_3_O_4_/GCE. The findings show that Co_3_O_4_ nanoparticles and RGO have a good synergy [62].Comparison of various electrode material for electrochemical detection of serotonin has been done in Table 5.

#### 5.2.4. Graphite and Carbon Fiber Electrode and Surface Modifications

Because of its strong electrically conductive characteristics, graphite is employed as an electrode material. Graphite, like metals, includes delocalized free electrons that can flow and conduct electricity. On the other hand, the basic carbon microfilament with a radius of micrometers is created with a tapered-tip insulating glass tube to form a carbon fiber microelectrode (CFM). In nature, CFMs are relatively innocuous. CFMs are useful for sensor applications due to their good physical and electrical qualities, which are a result of their high specific surface area and electric conductivity. Miyazaki et al. developed a novel carbon electrode with crystalline graphite powder supplemented by amorphous carbon, similar to pencil lead (GRC). With strong peak separation, this novel graphite electrode detects serotonin in the presence of DA, AA, and nitrite ions [64]. To acquire high levels of sensitivity, accuracy, selectivity, and repeatability, Wang et al. worked on a modified graphite electrode which was created by intercalating CNTs on the graphite surface and used to assess DA and ST concurrently in the presence of AA [65]. In other work by Kachoosangi and Compton, the sensor responds to AA at 0.07 V with a low DPV current response and ST at 0.32 V and DA at 0.13 V with a significant DPV current response. For the determination of ST, DA, and AA, an edge plane pyrolytic graphite electrode (EPPGE) outperformed many unmodified carbon-based electrodes, as well as several modified electrodes in terms of sensitivity, detection limit, and peak separation [66]. Gupta and Goyal employed polymelamine to improve the EPPGE. Melamine polymer (poly1,3,5-triazine-2,4,6-triamine) is a conducting polymer that provides reduced fouling, a change in peak potential to fewer positive values, and a modest improvement in detection limit [67] (Table 6).

Carbon paste is a mixture of an organic liquid and conductive graphite powder that can be used as an alternative to graphite electrodes. These electrodes are simple to make and provide an easily renewable electron exchange surface. A modified carbon paste electrode layered with iron(II)phthalocyanine (FePc) and iron(II)tetrasulfophthalocyanine ([FeTSPc]^4−^) was used to detect DA and ST, simultaneously, in the presence of interferent AA. The Fe(II)/Fe(III) redox pair mediates the electrocatalytic activity of these phthalocyanine complexes in the oxidation of serotonin [71]. Wei et al. explained that, due to little affinity for ionized species in the solution, the carbon paste electrode modified with a non-ionic Poly-2-amino-5-mercapto-thiadiazole film displayed better antifouling capabilities [72]. The polymer film demonstrated good repeatability and no interference between all three analytes as AA, DA, and ST. The sensors outperformed the graphite-based SMEs in terms of the detection limit; however, the ST detection range is insufficient. Furthermore, the sensing instrument was only evaluated with a pharmaceutical sample, not a real human serum sample. Babaei et al. used nafion, metal hydroxide nanoparticles, and MWCNTs to create a novel ionic liquid electrode. Because of the combined and simultaneous effect of all analytes, the surface having modifications demonstrated high selectivity for cationic neurotransmitters, ST, and L-Dopa. Interestingly with a high sensitivity in comparison to the conventional graphite-based electrodes at lower concentration ranges, the modified electrode provided a broader linear detection range for ST, despite the usage of Co(OH)_2_ NPs having no extraordinary effect on the detection limit [73] (Table 7). CFM has a longer track record than GCE in the study and field of electrochemical serotonin sensing and detection. Rivot et al. used a nafion-coated CFM in the rat’s spinal dorsal horn for in vitro and in vivo serotonin sensing [74], while Jackson et al. did the same experiment as conducted previously in the rat’s brain, using fast-scan cyclic voltammetry (Figure 8 and Figure 9) [75]. Other nafion-coated CFMs were reported with nafion preventing the electrode from being poisoned by brain tissue implantation. Irazu et al., in the year 2001, used nafion and electropolymerized films of [Ni(TSPc)] to modify CFM [76]. The carbon nanotubes pretreatments to the carbon fiber electrode improved selectivity, accuracy, sensitivity, reduced fouling, and improved electron transport for in vivo ST analysis in the striatum of anesthetized rats [77]. Carbon nanotubes, on the other hand, are unknown in terms of fouling reduction. Njagi et al. used a modified CFM with CNTs dispersed in nafion to track in-vivo serotonin levels in entire zebrafish embryos’ digestive systems in real-time. The range of linear detection having a lower limit was dramatically reduced when nafion was coated with CNTs or Ni phthalocyanine, compared to when nafion was used alone for CFM modification [78].

#### 5.2.5. t-ZrO_2_ NPs Modified CPE

The synthesis of ZrO_2_ NPs is performed using the gel-combustion method by combining zirconyl nitrate hexahydrate (ZNH) with glycine. To ensure the complete combustion of the materials, they were mixed in a ratio of 1:1. In a minimum amount of milli-Q-water, the ZNH and glycine as oxidizers and as fuel were mixed, respectively. The obtained transparent gel with high viscosity was transferred to a pre-heated muffle furnace at a temperature 500 °C, and the excess water from the redox mixture was evaporated. The complete combustion results in the formation of white voluminous, porous, and puffy mass used in various characterizations. The BCPE was formed by mixing graphite powder and silicone oil in an appropriate proportion of 30:70, similarly, ZrO_2_NPs/MCPE (ZMCPE i.e., zirconium oxide nanoparticles modified carbon paste electrode) was prepared ZrO_2_ NPs with graphite powder and silicone oil in the ratio of 10:10:80 by *w*/*w*/*w*. The CHI-660 c model of potentiostat was used for the differential pulse and cyclic voltammetry techniques. The conservative three-electrode cell system was made up of ZMCPE as the working electrode, and saturated calomel and platinum wire as a reference electrode and a counter electrode, respectively.

Potassium ferrocyanide [K_4_Fe(CN)_6_] is uniformly used to check the electrochemical properties of ZMCPE. The cyclic voltammetry of [K_4_Fe(CN)_6_] is conducted at ZMCPE with 0.1 mM concentration with 0.005 Vs^−1^ scan rate, and 1 M KCl as a supportive electrolyte showing low signals of current. The DPV experiment of ZMCPE shows a high sensitivity for serotonin quantification in the range 10,000–50,000 nM, and the limit of detection is 585 nM [80].

#### 5.2.6. Diamond, Screen Printed, and Metal or Metal Oxide-Based Electrodes with Surface Modification

Boron doped diamond electrodes (BDDE) provide a very low background current, a broad potential window, and chemical inertness. The in situ detection of serotonin was conducted using an unmodified BDDE. The performance of a high-quality, conductive BDD thin-film electrode for the detection of histamine and serotonin was demonstrated at the same time [81] and is unlike GCE, which is prone to fouling, deactivation, and deletion due to excessive adsorption of serotonin oxidation products. The discovery of serotonin oxidation was conducted on diamond electrodes with very minimal and reversible adsorption of the quinone oxidation product. The outcomes and performances of the three bare carbon-based electrodes, glassy carbon (GC), polycrystalline boron-doped diamond (pBDD), and carbon nanotube networks (CNTNs) were assessed for the precise and sensitive detection of serotonin. For pBDD, 10 nM for the CNTN electrode, and 2 M for GC, the detection limits for serotonin from CV were reported to be 500 nM, 10 nM for the CNTN electrode, and 2 M for GC. Liu et al. first investigated nafion membrane-coated colloidal gold screen-printed electrodes (SPE) for the amperometric sensing of serotonin in the blood and brain of depressed mice. In the presence of interferents, such as DA, AA, and UA with high concentrations, the colloidal Au-SPE modified with nafion membrane provided good outcomes with sensitive, rapid, accurate, and convenient sensing or detection of serotonin. By acting as a tiny conduction center in graphite/SPE, the gold has the ability in the colloidal form to improve the electron transfer rate and hence provide much higher sensitivity, increasing the rate of electron transfer [82].

The serotonin electrocatalytic response on the nafion/CGSPE, i.e., the nafion membrane-coated colloidal gold screen-printed electrode, was compared to bare SPE. The electrode with surface modifications had a higher serotonin selectivity and sensitivity. To detect ST and melatonin simultaneously, a carbon screen-printed electrode (CSPE) was modified with a range of carbon nanomaterials, such as SWCNTs, MWCNTs, or graphene [83]. Carbon nanomaterials added to the CSPE increased the electroactive area and, as a result, the electrocatalytic capabilities. However, due to a low detection limit, the results are not very promising. Y. Wang et al. described that the MWCNTs were also combined with ZnO and chitosan to create a new modified SPE-based sensor that could determine both NE and ST at the same time [84]. The composite mixture’s excellent electrocatalytic activity resulted in high current responsiveness and reduced LOD due to the combined advantage of all three components. Tertiş et al., (2017) added Au NPs and PPy NPs to the screen-printed electrode. According to the findings, hydrogen-bonding interactions between serotonin and the polypyrrole (PPy) polymer generate analyte buildup on the electrode surface. The effective intermolecular hydrogen bonding of DA and NE with the polymerized film is limited due to the attachment of two hydroxyl groups on the ortho position of the benzene ring in DA and NE. Because ST lacks intramolecular hydrogen bonding, it is possible to build up ST preferentially on the modified electrode surface. As a result, the current signal for ST increased with time, while the current signal for others decreased. In terms of sensitivity, the modified SPE outperformed the bare electrode by 320 times, emphasizing the benefits of this tailored hybrid surface [85].

Comparison of metal and screen printed electrodes for electrochemical serotonin detection has been done in Table 8.

### 5.3. Epinephrine

#### 5.3.1. CPE Modified with 3,4-Dihydroxybenzaldehyde-2,4-dinitrophenylhydrazone (DDP) and Carbon Nanotubes (CNTs)

The cyclic voltammetry, differential pulse voltammetry, and chronoamperometry tests were carried out by using PGSTAT-302 N Autolabpotentiostat with GPES software. The electrochemical cell consisted of three electrodes: a reference electrode consisting of a 3.0 M solution of Ag/AgCl/KCl, a platinum-wire counter electrode, and a modified DDP-CNPE as a working electrode. To detect the pH level a Metrohm pH-meter-691 was used and at room temperature, all trials are conducted with high accuracy and sensitivity. DDP-CNPEs were produced by hand in a mortar and pestle using 0.01 g DDP in ethyl alcohol, 0.1 g CNTs, and 0.89 g graphite powder. The above mixture was then mixed with 0.7 mL Paraffin for about 20 min until a consistently moist paste was created and put into the opening of a glass tube. To make an electrical contact, a Cu-wire was inserted into the prepared paste of carbon. The electroanalytical response of these new modified electrodes was examined for EP using CV, DPV, and chronoamperometry studies to describe them.

EP oxidation has a potential window value of around 215 mV which was slightly less positive than that of unmodified CPE at the optimal pH of 7.0. Two linear dynamic ranges of EP were observed using differential pulse voltammetry at the modified electrode, with a detection limit of 70 nM and concentration ranges of 100–35,000 and 35,000–750,000 nM for two linear segments with slopes of 0.019 and 0.152 µAµM^−1^. To confirm our findings, one mL of EP ampoule was diluted up to 10 mL in 0.1 M PBS at pH 7.0 and next, phosphate buffer was used to dilute different quantities of the diluted solution in a series of 10 mL volumetric flasks to the desired concentration. Each sample solution was placed in an electrochemical cell, and the DPV was monitored between 0.0 and 0.5 V at a rate of 10 mV/s. This method was conducted five to six times for each sample, yielding an average amount of EP in the injection of 0.989 mg, which was very close to the number on the ampoule label (1.0 mg) [93].

#### 5.3.2. Gold Electrode Modified with Polyaniline (PANI), MWCNT, RuO_2_, TiO_2_, MWCNT-PANI-TiO_2,_ and MWCNT-PANI-RuO_2_ Suspensions

After the synthesis of all types of modified suspensions, the procedure of electrode modifications started, the surface of the working electrode (Au) was polished using a SiC-emery paper in an aqueous slurry of alumina nano powder (LabChem, Zelienople, PA, USA). To remove any leftover alumina particles that may have been captured on the electrode’s surface, it was subjected to 5 min of ultrasonic vibration in distilled water and ethanol (100%). MWCNT-PANI-TiO_2_ suspension in DMF was made by combining MWCNT (5 g), PANI, and TiO_2_ (ratio 1:1:1) in DMF (1 mL) and sonicating for more than 24 h. Other nano-catalyst suspensions were made by dissolving 5 mg of chemically synthesized samples PANI, MWCNT, RuO_2_, or TiO_2_ in DMF (1 mL) and sonicating it for half an hour. The drop-dry approach was used to make modified electrodes of Au-MWCNT, Au-PANI, Au-RuO_2_, Au-TiO_2_, Au-MWCNT-PANI-TiO_2_, and Au-MWCNT-PANI-RuO_2_. All these PANI, MWCNT, RuO_2_, TiO_2_, MWCNT-PANI-TiO_2_, and MWCNT-PANI-RuO_2_ suspensions were dropped (5 µL drops) on the bare gold electrode and dried for 5 min at 50 °C temperature in a vacuum or simple oven and, after synthesizing these nanomaterials, characterization is done.

The goal of this research was to find the optimal electrode for improving electron transport qualities. The changes in CV (Figure 10) observed during the modification of electrodes with the produced nanomaterials suggest that MWCNT, PANI, TiO_2_, and RuO_2_ were successfully immobilized on the Au electrode surface. The [Fe(CN)_6_]^3−/4−^ redox process is assigned to the redox peaks observed near about 0.2 V; on the other hand, the oxidation peak is observed around 1.0 V on the modified Au-MWCNT-PANI-RuO_2_ which is attributed to Au^+^, to Au^3+^, or to Au^4+^. This 1.0 V peak was most likely not caused by the electrode’s quick electron transfer process. At the Au- MWCNT-PANI-TiO_2_ electrode, the redox couple of [Fe(CN)_6_]^4−^/[Fe(CN)_6_]^3−^ seemed reversible, with a peak current’s ratio of I_pa_/I_pc_ = 1.01 or nearly 1 for a non-irreversible process, and a peak separation of 154 mV, indicating a quick transfer of electrons. With I_pa_/I_pc_ = 1.0, however, Au-MWCNT-PANI-RuO_2_ looked to be quasi reversible. The composites (MWCNT-PANI-MO) modified electrodes had a larger peak current than the other electrodes, particularly the bare gold electrode. This is attributed to the enormous surface area produced by the MWCNT and PANI, which allows for facile electrolyte transport and easy electron flow due to the reduction and oxidation reaction of the TiO_2_ and RuO_2_ NPs at the electrode. 

At concentrations ranging from 4900 to 76,900 nM, a linear calibration plot was obtained for EP. The detection limits of EP for Au-MWCNT-PANI-RuO_2_ and Au-MWCNT-PANI-TiO_2_ electrodes were calculated to be 0.18 nM, and 0.16 nM, respectively. The interference analysis with DPV revealed that the amino acid and EP peaks were well separated. As a result, epinephrine can be detected by using modified electrodes without being hampered by AA signals. The analytical performance of the manufactured sensors was tested for epinephrine sensing and detection in a medicinal and pharmaceutical sample and found to be satisfactory [94].

#### 5.3.3. Gold Disk Electrode Using Biomimic Hemin Modified Molecular Imprinted Polymers (MIP)

Firstly, the functional monomer is synthesized for the preparation of EP imprinted and non−imprinted microspheres. A total of 20 mmol of melamine was mixed with 20 mL of dimethylformamide to make (TAT) 2,4,6-trisacrylamido-1,3,5-triazine, the chosen functional monomer. The reaction mixture was agitated for about 10 h after 65 mmol of AC was gently added to melamine in DMF. The leftover precursors were then washed out of the mixture with 10 mL DMF-DW (1:1 *v*/*v*) at 35 °C temperature. Using a non-covalent immobilization procedure, a hemin modified EP imprinted polymer (MIP) was created. In a 15 mL glass vial, 2 mmol TAT (functional monomer), 0.5 mmol EP (template), and a 5 mL mixture of DMSO and acetonitrile (3:2 *v*/*v*) were mixed to make the pre-polymerization complex. The components were vigorously agitated to facilitate host–guest chemistry to generate self-assembly. The materials 0.02 mmol hemin, 5 mmol ethylene glycol dimethyl acrylate (EGDMA cross-linker), and 0.1 g azobisisobutyronitrile (AIBN) were then added in that order. To establish an inert atmosphere, the mix solution was sonicated and cleansed with nitrogen gas for more than 10 min. To finish the polymerization reaction, the vial was packed, sealed, and maintained in a water bath for 15 h at 60 °C. The polymer was then rinsed with a lot of methanol to get rid of any leftover precursors. The resulting polymer was identified as microspheres, which were collected on a Nylon filter, with a pore size of 0.27 m and rinsed with the CH_3_OH-CH_3_COOH (9:1 *v*/*v*) mixture to eliminate the template. In the absence of a template, the reference or non-imprinted polymer was synthesized in the same way as EP. Finally, the MIP-modified electrode is prepared with several precautions. A gold disc electrode is further modified with molecularly imprinted microspheres supported by a chitosan/nafion combination to create an electrochemical sensor for EP detection. The suggested sensor has a linear concentration range of 50 nM to 40,000 nM, with an extremely low limit of detection, i.e., 12 nM (S/N = 3). Our findings show that an imprinted polymer has a more efficient sensing capability than other structural counterparts and potential interferants, with better reproducibility and selectivity for EP detection. Just because it contained two catalytic sites that aid in EP detection, the proposed electrochemical sensor essentially follows a cascade reaction process. Using human blood serum and injection samples, the analytical usefulness of this sensor for determining EP is demonstrated [95].

#### 5.3.4. GCE Modified with L-Glutamic Acid-Functionalized Graphene

On a CHI660A electrochemical workstation, electrochemical measurements were taken (Shanghai Chenhua Instruments Co., Shangai, China). A CHI 660D electrochemistry workstation was used to perform electrochemical impedance spectroscopy (EIS) experiments (Shanghai Chenhua Instruments Co., Shangai, China). In all of the studies, the working electrode was a modified GCE, while the reference and auxiliary electrodes were a saturated calomel and a platinum wire, respectively. A reaction between L-glutamic acid and graphene produced L-glutamic acid-graphene nanocomposites. In KOH solution (50 mL, 2 mg mL^−1^), 40 mg of L-glutamic acid and 10 mg of graphene oxide were added. After stirring for 23 h at 75 °C, 0.2 mol L^−1^ NaBH_4_ solution (6.0 mL) was mixed and swirled for two more hours. The L-glutamic acid-graphene hybrids were obtained by centrifuging the product and thoroughly washing it with ethanol and double-distilled water. Before being used, the GCE was burnished with alumina powder and cleaned thoroughly. The GCE was treated with 10 mL of L-glutamic acid-graphene aqueous solution (1 mg mL^−1^) and dried at room temperature. The L-glutamic acid-graphene/GCE electrode is the modified electrode. In the electrochemical behavior of the modified electrode for CVs of the bare GCE in [Fe(CN)6]^4−/3−^ solution(5 mM) containing 0.1 M KCl, a typical redox peak was detected. The redox peaks were significantly reduced, followed by the electrode exterior which was changed with graphene oxide to obtain graphene oxide/GCE, showing that the graphene oxide hampered electron transport. The modified L-glutamic acid electrode was made by sweeping the voltage from 0.8 to 1.7 V at 100 mV/s for ten cycles in phosphate buffer containing 5 × 10^−3^ M L-glutamic acid solution at pH 7.0, emanating in a reduction in peak current. In the CVs of L-glutamic acid-graphene/GCE, the redox peak was bigger than the redox peaks in graphene oxide/GCE. The resultant nanocomposite improved [Fe(CN)6]^4−/3−^ solution significantly, with an unambiguous redox peak between 0.2 and 0.6 V that was not evident in CV curve B.

The modified electrodes were utilized to evaluate the current response of epinephrine at varied doses using differential pulse voltammetry under optimum experimental conditions. It is observed that there was a linear connection between peak current and epinephrine concentrations between 1.0 × 10^2^ and 1.0 × 10^6^ nM (R = 0.9833) with a detection limit of 30 nM (S/N = 3). The analytical features of the L-glutamic acid-graphene modified electrode were compared to other epinephrine detection sensors to highlight the advantages. The outcomes are shown in Table 9. Although the detection limit for epinephrine in the sensor reported here was slightly higher than that of a graphene/gold/GCE [96], the sensor had a much higher current sensitivity and a lower detection limit for epinephrine with a wider linear dynamic range when compared to a graphene-modified electrode. The L-glutamic acid-graphene sensor outperforms previous sensors due to the synergistic consequence of the nanocomposite and the addition of L-glutamic acid [97].

### 5.4. Norepinephrine

#### 5.4.1. GCE Modified with CNTs and Cobalt Ferrite Magnetic Nanoparticles

The electrochemical analysis was proceeded using cyclic voltammetry, and chronoamperometry was carried out using an Autolab electrochemical system Galvanostat Model PGSTAT 30 with GPES software. A thermostatic electrochemical cell of 15 mL with three electrodes was used to make electrochemical measurements, 3.0 mol/L solution of Ag/AgCl/KCl), and a platinum wire was assembled as counter and reference electrodes, respectively, with a working electrode of area = 0.071 cm^2^. The cyclic voltammetric studies of norepinephrine via modified surface electrode were conducted by using 0.01 mol/L potassium ferrocyanide and potassium ferricyanide in 0.01 mol/L phosphate buffer of 7 pH, showing the potential range of 0.4 to 1.0 V against Ag/AgCl with a scan rate of 0.1 V/s. The chronoamperometry for noradrenaline or norepinephrine sensing and detection was carried out in 0.1 mol/L PBS with a fixed potential of 0.5 V against Ag/AgCl, with a stabilization period of 100 s. The GCE was polished and layered with 0.05 µm alumina solution before being rinsed with de-ionized water and put in an ultrasonic bathtub for about 5 min in ethyl alcohol, and 10 min in a water bath, before being modified. The prepared electrode was immersed in 0.1 mol/L H_2_SO_4_ solutions after electrochemical polishing, and cyclic voltammetry was used to measure the potential range of −1 to 1 V against the Ag/AgCl solution. Following that method, ten milligrams of MWCNT and ten milligrams of Fco_98_ were suspended in two milliliters of ethanol. For 30 min, an ultrasound was used to disperse the suspension.

The best composition of the electrode was 4 μL of cobalt ferrite and 10 μL of CNTs in 0.1 mol/L buffer solution of phosphate at pH 7.0. In 0.1 mol/L PBS, the electrode exhibits electrochemical activity across a broad potential range from 0.4 to 1.0 V against Ag/AgCl solution, showing excellent dynamism, electrode robustness, and stability. The original GC electrode was used to catalytically oxidize noradrenaline at +0.60 V opposite to the Ag/AgCl solution by fixing the current at 0.17 A, whereas the modified GCE containing cobalt ferrite NPs and CNTs was used with a solution of Ag/AgCl at +0.54 V with 0.23 mA of current. The concentration limit was 760 nM for the anodic peak current (Ipa) against the concentration of Noradrenaline, using the amperometry technique at the highly modified surface electrode which is linear in the concentration range from 160 to 1910 nM. In this approach, the modified electrode GC/MWCNT/Fco_98_ was discovered to have a precise and auspicious demand in the field of neuroscience for determining this neurotransmitter [100].

#### 5.4.2. GCE Modified with Graphene

For electrochemical measurements, the electrochemical workstation CHI 660 was employed, and the graphene was characterized using the Varian660-IR at Agilent Technologies, Santa Clara, CA, USA. A well-assembled three-electrode setup with f a bare or GCE/graphene with a3 mm diameter was used as the working electrode, while platinum wire was used as a counter electrode, and an electrode containing Ag/AgCl solution was used as a reference electrode.. After preparing the nano-graphene, the graphene-modified electrode was prepared as 7 mg of graphene and was disseminated at a concentration of 0.7 mg mL^−1^ in 10 mL of double-distilled water. Before each experiment, the GCE was layered and waxed with 0.05 m alumina powder and gold sandpaper. Between each polishing process, the layered electrode was cleaned thoroughly with double-distilled water before being immersed in 50% nitric acid, ethyl alcohol, and double de-ionized water in an ultrasonic bath and dried in the air. A total of 4 μL of graphene suspension was poured onto the GCE and drying it under an IR lamp, a graphene-modified electrode (GME) was created. The redesigned electrode had an outstanding electrocatalytic impact on the oxidation of norepinephrine in phosphate buffer solution (NE). This was also utilized to determine NE, UA, and AA by CV in the presence of epinephrine. With NE concentrations ranging from 600–120,000 nM and the LOD of 4000 nM, the decreased peak current showed a linear relationship. By detecting the reduced peak current in PBS pH 7.0, the modified electrode can be utilized to measure NE in the presence of UA, epinephrine, and AA. The method was successfully applied to assessing NE injection samples and has great sensitivity, selectivity, and stability [96].

#### 5.4.3. GCE Modified with MoO_3_ Nanowires (NWs)

For the typical synthesis of MoO_3_ nanowires, 0.7 g of (NH_4_)_2_MoO_4_ was properly dissolved in 40 mL double-distilled water with continuous moderate stirring. To maintain the pH at 1.0, a high concentration of HCl was put in drop by drop. After 30 min of stirring, the reaction mixture was placed in a teflon-lined autoclave followed by the hydrothermal reactions for varying lengths of time at 180 °C temperature. To eliminate any impurities or unreacted precursors, the final product was thoroughly washed numerous times with ethyl alcohol and deionized (DI) water. The product was well dried in an oven at 80 °C for more than 3 h, further described and manufactured as an electrode for the electrochemical oxidation of NE after it was ground. A Bio-Logic VMP-300 potentiostat was used to carry out the important electrochemical experiments using a standard three-electrode system consisting of an auxiliary electrode, reference electrode, and working electrode made by using platinum wire, Ag/AgCl solution, and GCE/modified or unmodified (3 mm of diameter), respectively. The GCE was properly and gently polished with a 0.05 m alumina slurry and properly rinsed with distilled water before alteration. To remove any adsorbed alumina particles, de-ionized water, and 100% ethyl alcohol were used to wash the electrode carefully. The MoO_3_ NWs were dispersed in a 0.2% nafion solution in 1 mL. After that, these suspensions in a very small amount of approximately 20 µL were dropped on the pure and clean GCE, then allowed to dry at 25 °C or room temperature. After that, all electrodes were cleaned two or three times with de-ionized water before being employed in the selective recognition and binding studies. For all the electrochemical studies, the potential scanning was repeated in a range between 0.0 to 0.8 V, fixing the scan rate at 30 mV/s. The MoO_3_ NWs were made using a one-step hydrothermal process. The pH of the reaction medium affects the development of high-aspect-ratio nanowires during synthesis. 1-dimensional MoO_3_ NWs are generated in the first stage as follows:Mo_7_O_24_^6−^ + 6H^+^ + 11H_2_O → 7MoO_3_.2H_2_O → MoO_3_.H_2_O + H_2_O MoO_3_.H_2_O → MoO_3_ + H_2_O

The impact of reaction time on NW growth was also investigated, and all other synthesis settings were kept constant. For sophisticated biosensor devices, it is quite critical to achieving the direct transfer of electrons between a customized electrode and a biomolecule. In a systematic growth evolution investigation, 1-dimensional (1D) MoO_3_ nanowires (NWs) were created. These MoO_3_ NWs were also employed as a mediatorless biosensor electrode with glassy carbon electrodes (GCEs) for the sensing and accurate detection of NE using cyclic voltammetric techniques. In the electrochemical detection of NE, the GCE/MoO_3_ NWs exhibited a fantastic response time of 2 s and a LOD of 0.11 nM (Figure 11). The 1D microstructure and high catalytic activity provide a channel for the transport of electrons and increase their sensitivity, resulting in superior bio-electrochemical performance [101].

#### 5.4.4. GCE Modified with Cetyltrimethylammonium Bromide Assisted SnO_2_ Nanoparticles for Concurrent Sensing of Epinephrine and Norepinephrine

For the simultaneous sensing of epinephrine and norepinephrine, a highly selective, accurate, and sensitive electrochemical sensor was developed by using SnO_2_ NPs, assisted by cetyltrimethylammonium bromide (CTAB). Electrochemical measurements proceeded using a CH Electrochemical Workstation Instrument CHI-900 (Bee Cave, TX, USA), which features an electrode system of three electrodes, including a well-assembled GCE working electrode, wire of platinum, and 3 M KCl solution of Ag/AgCl as an auxiliary and reference electrode, respectively. The measurements were carried out by using SWV in a buffer solution of Phosphate at pH 5.0, including NEP and EP at an amplitude of 50 mV, frequency of 10 Hz throughout the potential range from 0.6 to −0.4 V with the step potential of 5 mV.

The behavior showed by EP and NEP in electrochemical studies at different GCE/SNO_2_ nanoparticles of modified electrodes was examined using the SWV method using buffer solution of phosphate at 5.0 pH to observe reduction electrocatalytically. According to the figure, the curve A of Bare GCE and SnO_2_/GCE, the two neurotransmitters, had a relatively low current response as in curve B. By observing curve C, when the CTAB-SnO_2_/GCE was utilized, the combination showed two excellent, sharp, clear, accurate, and sensitive peaks corresponding to NEP and EP at 0.309 V and −0.045 V results in a high separation of the peak in PBS of pH 5.0 with value equals to 0.354 V. For NEP and EP, the curve of calibrations was achieved under ideal conditions throughout a large concentration range of 0.1–300 and 0.1–250 M, respectively, showing the lowest values of detection limits i.e., 6 nM and 10 nM. The outstanding activity and selectivity were revealed by interference experiments for the modified electrodes in the presence of interferents viz. ascorbic and uric acid. The proposed modified electrode has been auspiciously used to sense NEP and EP, simultaneously, in urine samples of human beings with satisfactory results [102].

Comparison of various modified GCE for electrochemical detection of norepinephrine in Table 10. 

#### 5.4.5. Bare Boron-Doped Diamond Electrode

The electrochemical analysis was studied using an Autolab-Potentiostat of Type 3 (Metrohm Autolab B.V. Utrecht, The Netherlands) and 4.9 GPES software Utrecht, The Netherlands. The raw SWV signals observed and obtained by the electrochemical instrument were recorded in this software after being corrected using the moving average approach, with a peak width of 0.03 V and smoothed using an algorithm of the Golay and Savicky. All the voltammetric measurements were carried out with a system with three electrodes at room temperature using a 10 mL glass electrochemical Cell. The counter and reference electrodes were made of wire of platinum and a 3 mol/L solution of Ag/AgCl.

Using the cyclic voltammetry approach, NE showed one clear well-defined, adsorption controlled, and irreversible oxidation peak at about +0.85 V against Ag/AgCl solution electrode in 0.1 M Britton–Robinson buffer (BR) fixing pH at 2.0. On the oxidation peak current, the influence of supporting electrolyte, solution pH, and instrumental variables were optimized. In BR buffer solution using the square-wave stripping mode (SWSM), it was discovered that an excellent connection was developed between the concentration of NE and oxidation peak current in the range of 1–100 μg m/L (4900–490,000 nM) with a detection limit value of 0.254 μg m/L i.e., 1200 nM. This study effectively shows that the bare BDD electrode, a state-of-the-art electrode material, may be used as an electrochemical sensor to determine NE in pharmaceutical formulations, without the need for any chemical modifications. Furthermore, the NE concentration in the commercial pharmaceutical formulation is statistically equivalent to the NE concentration obtained using the HPLC method as the reference technique. Because it is relatively easy, economical, and quick, the combination of SWV and oxygen-terminated OT-BDD is a handy and advantageous alternative for determining NE in pharmaceutical formulations [107].

Comparison of norepinephrine detection using various electrodes are compiled in Table 11.

#### 5.4.6. CPE Modified with Graphene-QDs/ILs for Simultaneous Detection of Acetylcholine and Norepinephrine

An Autolabpotentiostat/galvanostat was used to perform the electrochemical experiments. The GPES software was used to regulate the experimental settings. At 25 °C temperature, a standard system of the three-electrode cell was employed. The reference, auxiliary, and working electrodes were 0.3 M Ag/AgCl/KCl solution, platinum-wire, and GQDs/2CBF/CPE/IL, respectively. pH measurements were taken with a Metrohm 710 pH meter. GQDs/IL/CPEs were made by using a mortar and pestle to combine graphene quantum dots (0.2 g), graphite powder (0.8 g), and ionic liquids of about 0.8 mL. After that, the paste was stuffed into the glass tube and the electrical circuit or contact was made using a Cu-wire put into the carbon paste. The samples of urine were refrigerated immediately after collection. Centrifugation of 10 mL of a sample at 2000 RPM for fifteen minutes was done. A 0.45 m filter was used to filter the supernatant. The prepared solution was then shifted to 25 mL of volumetric flask and diluted to the desired concentration with PBS at pH = 7.0. Different doses of norepinephrine and acetylcholine were used to anesthetize the diluted urine samples. The suggested process used the usual addition method to determine the content of norepinephrine and acetylcholine.

CV and DPV in a phosphate buffer solution were used to examine the electrochemical behavior of norepinephrine at GQDs/IL/CPE (pH 7.0). As an electrochemical sensor, GQDs/IL/CPE showed catalytic activity in the oxidation of norepinephrine. On the modified electrode, the potential of norepinephrine oxidation was shifted to more negative potentials, and the oxidation peak current increased. Furthermore, at E = 390 mV, the GQDs/IL/CPE presents two independent oxidation signals, enabling the simultaneous study of norepinephrine and acetylcholine. The obtained data revealed a solid linear association between norepinephrine and acetylcholine oxidation peak currents and their concentrations in the range of 20–400,000 nM. Finally, GQDs/IL/CPE has good accuracy and precision in determining norepinephrine (LOD = 60 nM) and acetylcholine in norepinephrine ampoule, acetylcholine ampoule, and urine samples [115]. Fajardo et al., (2019) explained the sensing of NE using a GCE modified with graphene quantum dots and gold nanoparticles. By using the SWSV technique, a low value detection limit was measured in the rat brain tissue [116]. In another work by Chen et al., a novel composite of MIP coated pd-NPs was synthesized by the sol-gel method. The combination of pd-NPS and silica-based MIP endowed the composite with good electrocatalytic chemical property, template selectivity, and a large surface area [117].

Comparison of electrochemical norepinephrine detection at various modified electrodes is compiled in Table 12.

## 6. Conclusions and Future Research

In this review, we highlighted the most recent advancements in electrochemical sensors for the detection of monoamine NTs in real samples at a nanomolar scale. Because these hydroxyl group-based monoamines are created locally in a specific region of the brain, the number of biomolecules is also present in the extracellular space. The precise quantification of NTs in vivo is difficult due to the ultra-low concentrations of NTs (nM). To overcome this problem, various kinds of modified electrodes are formed, used, and explained to determine the sensitivity of NTs. Because of their exceptional features, such as high catalytic activities, electron transfer rates, and electrical conductivity, the diverse carbon, metal/metal oxides, and polymeric nanocomposite sensing materials are incorporated into macro or microelectrodes for quick detection of NTs. The unique advantages of microelectrodes over the traditional electrochemical sensors are as they are reflected in the study of the active mechanism of electrochemical sites: the modulation strategies are more precise and controllable; they provide in situ analysis and characterization; and the microelectrode’s design allows for a more efficient reduction in the theoretical model; and they allow for a more consistent confirmation of the theoretical, microscopic, and macroscopic levels. Diverse modern materials, such as carbon derivatives, metal, and polymeric nanocomposite are used as electrode materials, and they are also used as electrode modifiers. Carbon derivatives, such as graphite, graphene, fullerene, and diamond electrodes are commonly used in sensing applications. In general, carbon materials are preferred due to their wide potential window, cost-effectiveness, and biocompatible properties. Still, they have disadvantages, such as higher ET resistance when compared to metals. Metals-based modifiers are not biocompatible and are hazardous. Therefore, there is a huge scope for nanocomposite electrodes to overcome these disabilities; however, cost and tedious preparation and characterization of nanocomposites are still major concerns. These sensing materials recently received a lot of attention and give a very low limit of detection with a precise linear range. Future research will focus on developing high-throughput miniaturized probe arrays using micro/nanofabrication techniques and integrating nanostructured sensing materials to improve long-term stability, sensitivity, selectivity, and reproducibility for rapid diagnostic and point-of-care detection of NTs in real-time. We emphasized the significant progress made in the field of electrochemical sensing for in vivo neurotransmitters through this review. When it comes to diagnostic gadgets, however, there are still certain obstacles to overcome. For starters, probe size is still an issue. Implantation causes a physical change in the brain’s tissue. To limit injury and inflammation, ongoing efforts to produce probe diameters comparable to interneuron lengths are being made. Furthermore, synaptic measurements will be possible due to continuous downsizing to nanoscale sizes. Second, electrochemical sensing can only detect a limited number of electroactive substances. Electrochemical sensors cannot detect peptides and proteins; thus, they cannot be measured with the same spatiotemporal resolution as smaller, more traditional neuromodulators. Modifications to electrodes using aptamers could be on the horizon for this type of research in the future. Finally, and maybe most importantly, there are no clinical measurements in humans. However, because of the tremendous progress in the field described here, we are certain that such measurements will be made possible soon.

## Figures and Tables

**Figure 1 materials-15-05782-f001:**
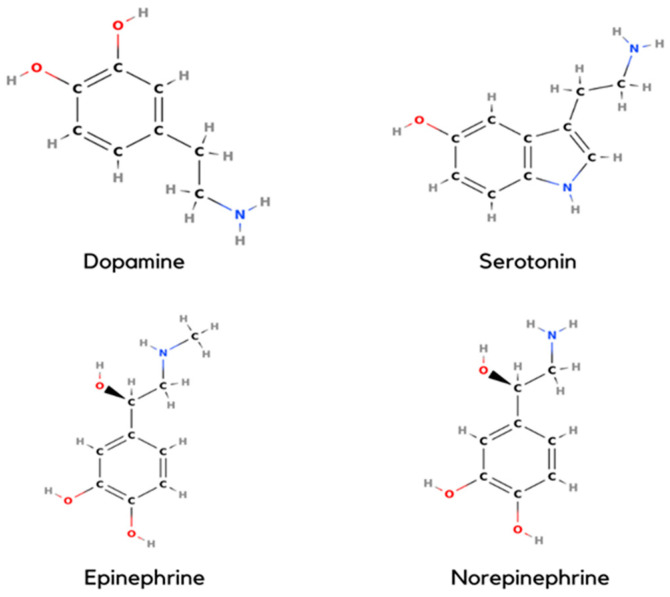
Structural comparison on the basis of position of hydroxyl groups.

**Figure 2 materials-15-05782-f002:**
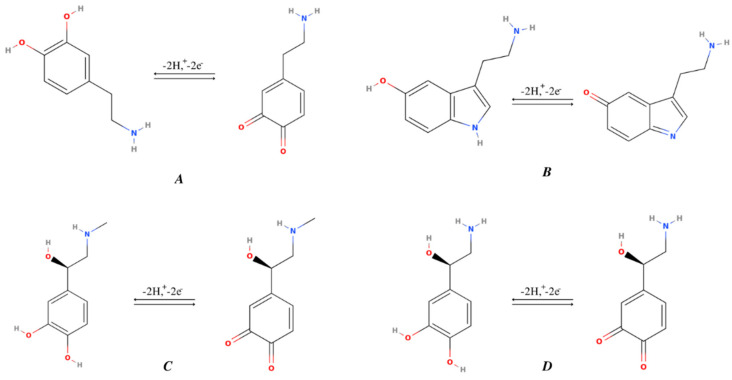
General oxidation mechanism of (**A**) dopamine (**B**) serotonin (**C**) epinephrine and (**D**) norepinephrine.

**Figure 3 materials-15-05782-f003:**
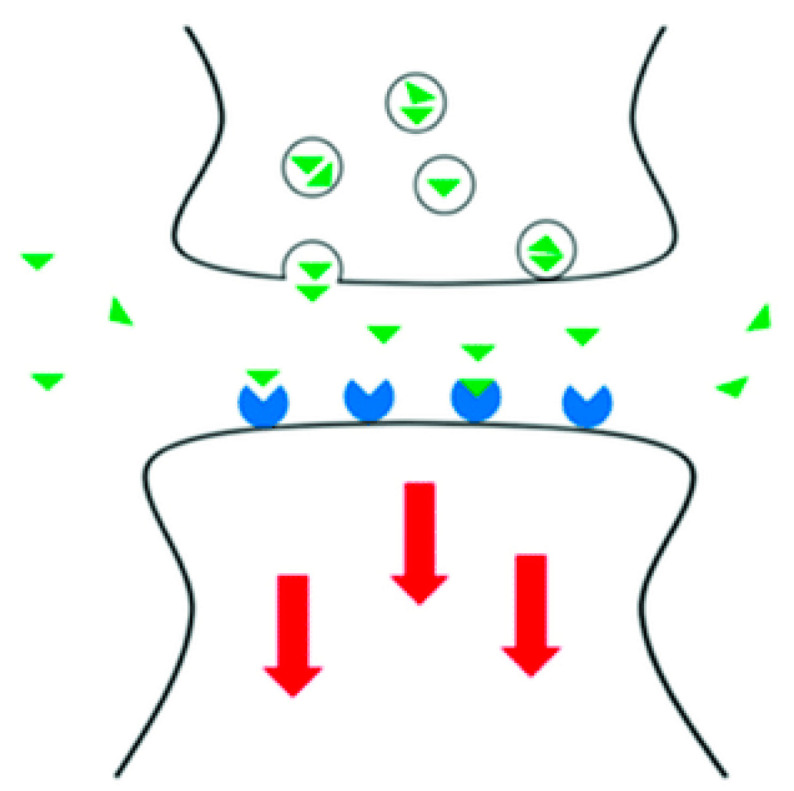
Neurotransmitters (green) are exocytotically released into the synaptic cleft and diffuse into the extracellular matrix, bind to the receptors (blue), and trigger a series of downstream reactions (red) in the axon of other neurons. Adapted from [15].

**Figure 4 materials-15-05782-f004:**
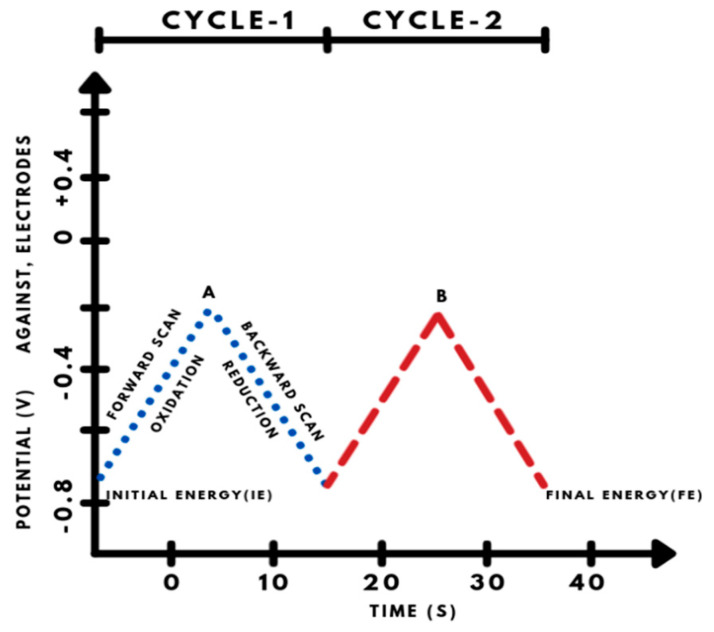
A typical triangular waveform used in cyclic voltammetry.

**Figure 5 materials-15-05782-f005:**
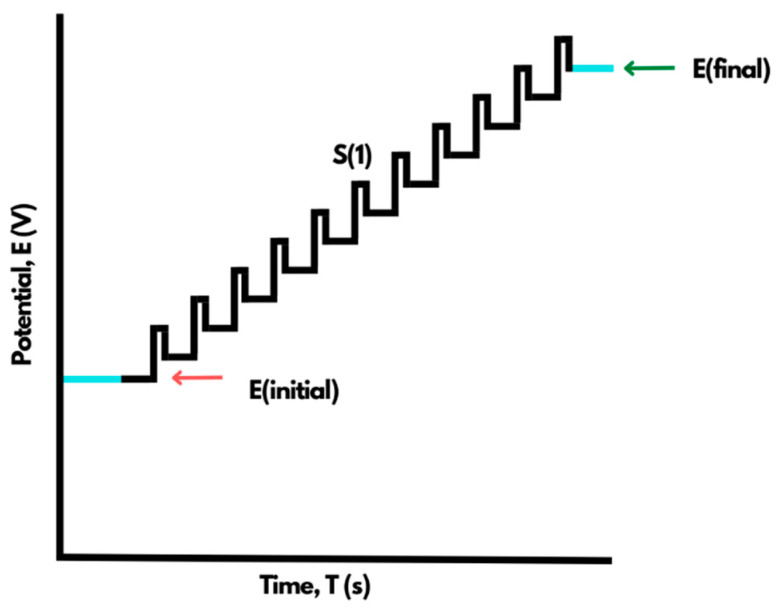
Diagram of pulses in the differential pulse voltammetry (DPV) technique.

**Figure 6 materials-15-05782-f006:**
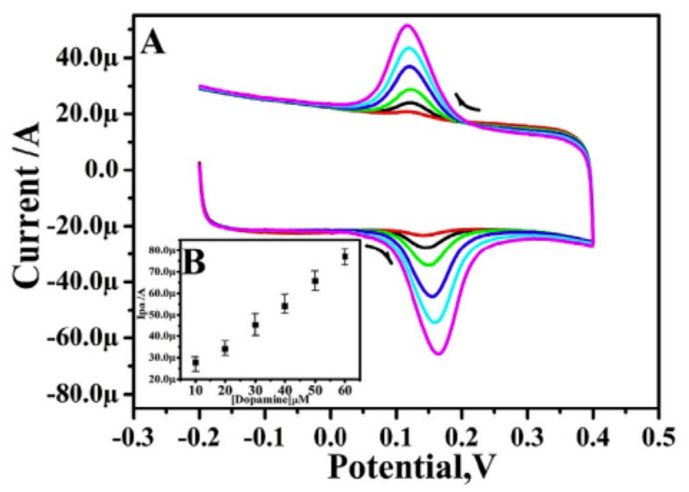
(**A**) Cyclic voltammogram of dopamine at different concentrations with scan rate of 100 mV/s in 0.2 M PBS solution of 7.4 pH at MCPE. (**B**) Graph of dopamine concentration versus current of anodic peak. Adapted from [28].

**Figure 7 materials-15-05782-f007:**
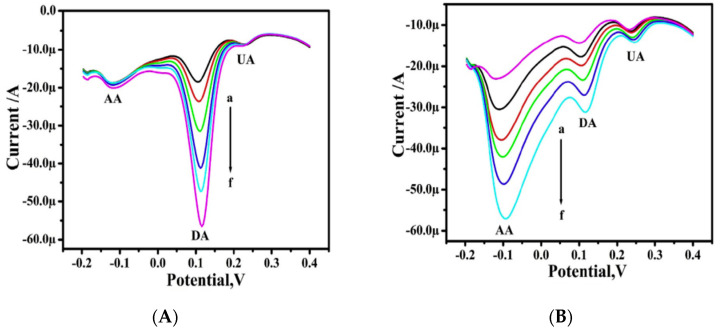
(**A**) DPVs were obtained at poly (Reactive blue) MCPE for concentrations of 5000–30,000 nM (a–f) DA in PBS at pH 7.4 in the presence of 5000 nM UA and 1 mM AA. (**B**) At poly (Reactive blue) MCPE, DPVs were produced for a range of concentrations of 1-6 mM (a–f) AA in PBS at pH 7.4 in the presence of 5000 nM DA and UA. Adapted from [28].

**Figure 8 materials-15-05782-f008:**
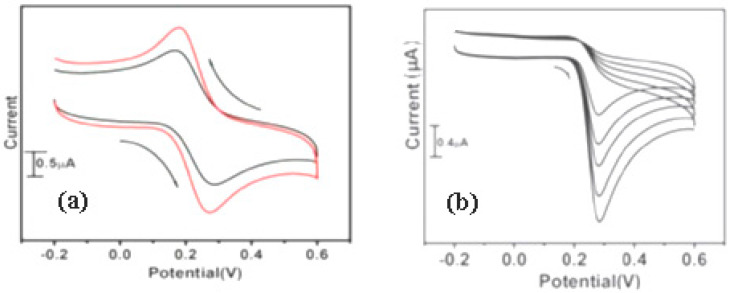
(**a**) CV of 10,000 nM serotonin in at BCPE (black line) and ZMCPE (red line). (**b**) CV of serotonin at ZMCPE with different concentration (10,000 nM to 60,000 nM) (0.2 M PBS pH 7.4 and scan rate of 0.05 V/s). Adapted from [75].

**Figure 9 materials-15-05782-f009:**
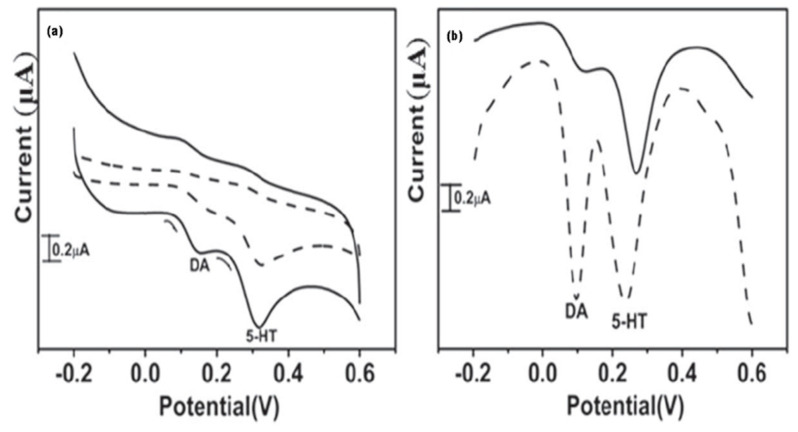
Interference study of dopamine in presence of serotonin by using (**a**) CV and (**b**) DPV technique done at ZMCPE having 0.1 mM concentration with 0.005 Vs^−1^ scan rate at pH 7.4. Adapted from [75].

**Figure 10 materials-15-05782-f010:**
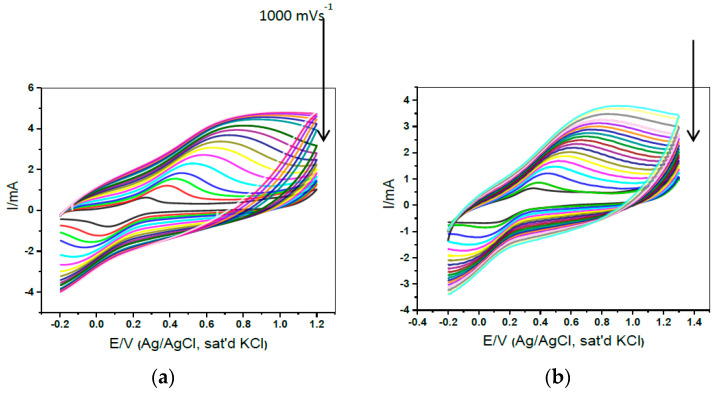
Cyclic voltammetry evolutions of (**a**) Au-MWCNT-PANI-TiO2 and (**b**) Au-MWCNTPANI-RuO_2_ electrode obtained in 5 mM [Fe(CN)_6_]^4−^/[Fe(CN)_6_]^3−^ (scan rate range 25–1000 mVs^−1^; inner to outer [94]. Here the black arrows represents sweep current.

**Figure 11 materials-15-05782-f011:**
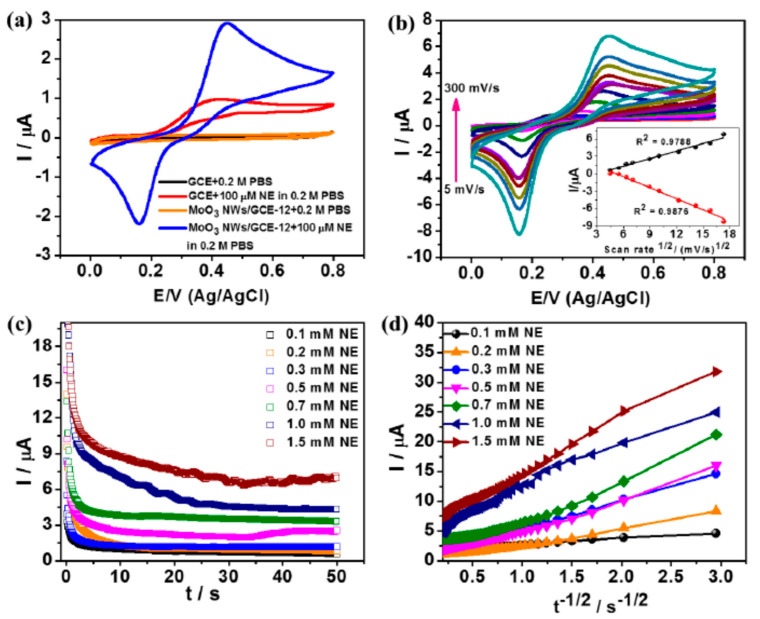
(**a**) CV curve of GCE and MoO_3_ NWs/GCE, with and without 100 µM NE in 0.2 M PBS at a scan rate of 30 mV/s. (**b**) Effect of scan rate from 5 mV/s to 300 mV/s in 0.2 M PBS. Inset of (**b**) is plot of Ip vs. υ^1/2^. (**c**) CA measurements of MoO_3_ NWs/GCE in the presence of various NE concentrations in 0.2 M PBS at 0.4 V. (**d**) Plot of I vs. t^−1/2^ [101].

**Table 1 materials-15-05782-t001:** General comparison of monoamine neurotransmitters. (ELISA—Enzyme Linked Immunosorbent Assay; HPLC—High Performance Liquid Chromatography).

Name	Localization	General Detection Method	Functions
Dopamine	Central and peripheral nervous system	ELISA and HPLC	Regulation of trennel, hormonal, and CNS, neuromodulatory
Serotonin	Central and peripheral nervous system	~ELISA	Embryo-genesis and morphogenesis
Epinephrine	Sympathetic nerves and adrenal medulla	Gas-liquid chromatography	Heart, muscle contraction, dilation of airways, glycogenolysis
Norepinephrine	Sympathetic nerves and adrenal medulla	Gas-liquid chromatography	Same as above

**Table 2 materials-15-05782-t002:** Comparison of DA sensing using CV and DPV with various electrodes modified with different materials.

S. No.	Electrode	Method	pH	Range (nM)	LOD (nM)	Sensitivity (µAµM^−1^)	Reference
1.	CAT/ZnONPs/CPE	CV	7	5000–41,000	3000	0.42	[31]
2.	TiO_2_ NP/CPE	SWV	7	1000–6000	840	0.042, 0.043	[32]
3.	Rod-shaped CuO NP/MCPE	DPV	6	300–1400	180	0.29	[33]
4.	INP-Nafion-modified CPE	DPV	7	10,000–110,000	3300	0.16	[34]
5.	GS/DMF/GCE	DPV	7.4	4000–100,000	2640	0.07	[35]
6.	Nanostructured Gold	DPV	7	10,000–100,000	5000	0.14	[36]
7.	CuO/GCE	CA	7.4	5000–40,000	100	0	[37]
8.	MIPs/CuO/GCE	CV	7.5	20–25,000	10	0.27	[38]
9.	CuO Nanoleaf	DPV	8	1000–1750	500	0.0229, 0.664	[39]
10.	S-β-CDCPE	CV DPV	7	500–50,000	13,300		[25]
11.	CuO nano rice/GCE	DPV	7	1000–150,000	420	0.04	[40]
12z.	PAYR/CPE	CV DPV	7	490–70,100	160		[24]
13.	Cu/Cu_x_O NPs/PGE	DPV	5.8	300–53,000	1070	0.51	[27]
14.	poly(XO)/GPE	CV DPV	7	-	9100		[26]
15.	MCPE-poly (Reactive blue)	CV DPV	4	5000–30,000	4600	-	[28]
16.	CTAB/CPE	CV	7	1000–70,000	200	-	[30]

**Table 3 materials-15-05782-t003:** Comparison of serotonin detection using CV and DPV based on GCE modified with polyelectrolytes and polymers.

S. No.	Coating	Technique Used	LOD (nM)	Range (nM)	Interference	Sample	Reference
1.	C-undecyl Calix resorcinarene	DPV CV CC	30	100–10,000	DA AA FA EP	Human serum	[41]
2.	Nafion/RuO_2_ pyrochlore	SWV	2	10–500 2000–20,000	UA AA glucose oxalate	Human blood Plasma	[42]
3.	Ch or Ach	CV DPV	500	1000–30,000	DA AA	Human plasma	[43]
4.	5-HTP	DPV	1700	5000–35,000	DA	Human serum	[44]
5.	DTDB	CV DPV	50	200–10,000	DA AA KCl NaCl UA Ca Glucose	Serum	[45]
6.	Poly(phenosafranine)	DPV	5	-	AA DA	Serum	[46]
7.	Eriochrome Cyanine R	CV DPV	50	50–5000	NE AA UA	Serum	[47]
8.	Poly(safranine O)	CV DPV	5	30–10,000	DA AA UA bilirubin TOC glucose	Serum	[44]

**Table 4 materials-15-05782-t004:** Comparison of serotonin detection using CV, and DPV based on GCE modified with carbon nanomaterials.

S. No.	Coating	Technique Used	LOD (nM)	Range (nm)	Interference	Sample	Reference
1.	MWCNT/DHP	CV DPV	5	20–5000	DA AA UA	Human Serum	[3]
2.	SWCNT	SWV	32	100–100,000	HIAA DA UA AA	Human urine	[56]
3.	MWCNT/IL gel	CV DPV	8	20–7000	AA glucose NaCl UA purine His Cys	Human serum	[50]
4.	MWCNT/CHT	CV DPV CA	80	500–130,000	L-Dopa AA Trp Tyr Ala glutamic acid His aspartic acid UA OA glucose Cys	Human serum and urine	[53]
5.	MWCNT/CHT	CV DPV	50	50–16,000	AA DA UA	Blood serum	[51]
6.	MWCNT/CHT/Poly (p-ABSA)	DPV	80	100–100,000	AA DA UA	Artificial urine	[52]
7.	Nafion/Ni(OH)_2_/MWCNT	CV DPV CA	3	8–10,000	DOPAC citric acid GC glucose NaCl KCl MgCl_2_ CaCl_2_ Ca(NO_3_)_2_ oxalate	Human serum	[53]
8.	Reduced graphene nano-sheets	CV DPV CA	32	1000–100,000	AA DA	Human serum	[54]
9.	G-g-PLA-Pd	CV Amp	80	100–100,000	UA DA AA H_2_O	Human serum	[55]
10.	Porphyrin derivative-GO	DPV Amp	4.9	100–300,000	UA AA Na^+^ Mg^2+^ Zn^2+^ Fe^2+^ Fe^3+^ Cl^−^	Human serum	[4]

**Table 5 materials-15-05782-t005:** Comparison of serotonin detection using CV and DPV based on GCE modified with metal or metal oxide nanoparticles.

S. No.	Coating	Technique Used	LOD (nM)	Range (nM)	Interference	Sample	Reference
1.	RGO/Co_3_O_4_—nanocomposite	CV DPV	48.7	100–51,000	DA AA EP UA Trp Cys vitamin B 6 vitamin B 9 Tyr guanine-cytosine NO_2_ NO_3_ KCl	Human serum	[62]
2.	PEDOT/RGO/Ag-hybrid nanocomposite	CV DPV CA	0.1	1–500,000	UA AA Tyr L -Cys L -Trp L -Ala DA Ep NE glucose	BAM solution	[60]
3.	MWCNT doped with nickel, zinc and iron oxide NPs	CV SWV	118 (NI)	5.9–62,800	DA AA	Human urine	[63]
4.	Fe_3_O_4_ NPs–MWCNT–poly BCG)	DPV	80	500–100,000	DA UA AD Trp L -Cys glucose	Human serum	[57]
5.	DDF-CNTs-TiO_2_/IL	CV DPV	154	1000–650,000	IP Cl^−^ F^−^ Br^−^ S^2−^ CO_3_^2−^ K^+^ Ca^2+^ Mg^2+^ NH^4+^ NO_3_^−^ CH_3_COO^−^ L-lysine glucose fructose sucrose L-asparagines GA L-Cys L-Trp AA UA D-penicilamine	Human serum	[33]
6.	RGO/PANI/AuNPs/MIPs	CV DPV	11.7	200–10,000	AA DA UA EP	Human serum	[60]
7.	L-Cys/AuNPs	CV CC SWV	20	60–6000	EP DA AA FA	Human serum	[59]
8.	Nano-Au/oxidized-PPyox	DPV	1	70–2200	DOPAC UA AA DA Glucose oxalate KCl MgCl_2_ NaCl CaCl_2_ Ca(NO_3_)_2_	Human serum	[64]

**Table 6 materials-15-05782-t006:** Comparison of serotonin detection using CV and DPV based on Graphite electrodes with modifications.

S. No.	Bare Electrode	Coating	Technique Used	LOD (nM)	Range (nM)	Interference	Sample	Reference
1.	Graphite	Intercalating CNT	CV DPV	200	1000–15,000	DOPAC AA DA UA oxalate glucose	Rabbit brain homogenate	[68]
2.	EPPGE	No coating	CV DPV	60	100–100,000	DA AA	Horse blood	[66]
3.	EPPGE	Poly-melamine	CV SWV	30	100–100,000	AA xanthine DA hypoxanthine	Human serum and urine	[67]
4.	PGE	Poly(pyrrole−3-carboxylic acid)	CV DPV ADSV	2.5	10–1000	DA AA UA NaCl KCl Urea Citric acid TA Glucose	Blood serum and urine	[69]
5.	CPE	Non-ionic PAMT film	CV DPV Amp	0.4	20–1560	AA DA Na^+^ K^+^ Mg^2+^ CH_3_ CO^−^ SO_4_^2−^ Ca^2+^ Cl^−^ PO_4_^3−^ BO_3_^3−^	Pharmaceutical samples	[58]
6.	CILE	Nafion/Co(OH)_2_- MWCNTs	CV DPV	23	50–75,000	L-Dopa DOPAC citric acid UA AA Glucose GA NaCl Ca(NO_3_)_2_ CaCl_2_	Human serum	[53]
7.	Graphite	Polyrutin modified paraffin	CV DPV	100	300–9000	EP AA	-	[70]

**Table 7 materials-15-05782-t007:** Comparison of serotonin detection using CV and DPV based on Carbon fiber electrodes with modifications.

S. No.	Coating	Technique Used	LOD (nM)	Range (µM)	Interference	Sample	Reference
1.	Nafion	FSCV	1	1000–100,000	-	Rat brain (in vivo)	[78]
2.	PPyox/CT-DNA	CV DPV	7	10–1000	AA DE DOPAC UA glucose oxalate	Human serum	[79]
3.	CNTs/Nafion	DPV	1	5–200	-	Zebra-fish (in vivo)	[78]
4.	SWCNTs	FSCV	-	-	DA	Striatum of rat	[77]
5.	Au-NPs/CT-DNA	CV DPV	800	80–200,000	DA AA glucose-fructose purine citric acid GA NH_4_NO_3_	-	[68]

**Table 8 materials-15-05782-t008:** Comparison of serotonin detection using mainly CV and DPV based on screen printed and metal-based electrodes with modifications.

S. No.	Electrode	Coating	Technique Used	LOD (nM)	Range (µM)	Interference	Sample	Reference
1.	ITO	Gold NPs	DPV SWV	3	10–250,000	DA AA UA NaCl KCl caffeine TA	Human serum and urine	[86]
2.	ITO	Ag ion irradiated MWCNTs	CV SWV	0.75	10–105,000	AA DA UA	Human serum and urine	[87]
3.	FTO	PEDOT: PSS/ 3IP-TPyP	CV	230	1700–138,000	DA EP AA Urea, D- Glucose L-Dopa	-	[88]
4.	ITO	AuAg-GR nanohybrid	CV Amp	1.6	2.7–4820	UA AA Glucose KCl	Human serum	[89]
5.	Acupuncture needle	PEDOT/CNTs	DPV Amp	78	1000–100,000	His bradykinin AA UA Prostaglandin E 2	Rat (in vivo)	[90]
6.	Pt	MWCNT/PPy/ Colloidal AgNPs	DPV CA	150	500–5000	AA UA	Artificial plasmatic serum	[91]
7.	Gold	Captopril/ thiophenol	DPV EIS	28	4000–250,000 2–3500	AA DA Trp Glutamate 5-HTP NE ACh Asp Gly Tryptamine His Adenosine	Human serum	[92]
8.	Carbon-SPE	SWCNTs or MWCNTs or Graphene	DPV	400	1000–2,500,000	UA MEL Tyramine Tryptamine Ep	Commercial drugs	[83]
9.	Graphite based SPE	PPy NPs-Au NPs	CV SWV	33.22	100–15,000	DA NE Paracetamol AA Acetylsalicylic acid Glucose	Human serum	[85]
10.	CGSPE	Nafion	CV DPV	5	20–500,000	UA DA AA Glucose NaCl TA oxalate	Brain homogenate and platelet-rich plasma of mice	[82]
11.	SPE	MWCNTs-ZnO/CHT composites	CV SWV	10	50–1000	AA UA NE glucose Ca Mg citric acid	Rat cerebrospinal fluid	[84]
12.	CP	ZMCPE	CV DPV	585	10,000–50,000	DA	Real samples	[80]

**Table 9 materials-15-05782-t009:** Comparison of epinephrine detection using CV and DPV, based on various electrodes with modifications.

S. No.	Electrode	Technique Used	pH	LOD (nM)	Range (nM)	Sensitivity (mA-mM^−1^)	Sample	Reference
1.	Modified CNT carbon paste electrode	DPV	7	70	100–35,000 35,000–750,000	0.019 0.152	-	[93]
2.	MWCNT-PANI-TiO_2_	DPV	7	0.16	4900–76,900	785.7	Pharmaceutical sample	[94]
3.	MWCNT-PANI-RuO_2_	DPV	7	0.18		79,226	-	[94]
5.	Electropolymerized MIP	Amp	-	9	300–100,000	-	-	[96]
6.	Hemin modified MIP	DPV	7	12	50–40,000	-	Human blood serum and injection samples	[95]
7.	PLG-Ag/GCE	CV DPV	7.5	800	3000–100,000	-	-	[98]
8.	L-GlutamicAcid-Graphene/GCE	DPV CV	4	30	100–1,000,000	0.2	Human urine	[97]
9.	TiO_2_/RGO-MCPE	CV DPV	7	3	5–100 100–2000	-	-	[99]

**Table 10 materials-15-05782-t010:** Comparison of norepinephrine detection using mainly CV and DPV based on glassy carbon electrodes with modifications.

S. No.	Electrode	Technique Used	pH	LOD (nM)	Range (nM)	Sample	Reference
1.	GR/Pd/GCE	CV DPV	7.2	0.064	5000–500,000	Pharmaceutical dosage forms and human urine	[103]
2.	DMSA/Au/GE	CV	7	0.055	100–700,000	-	[104]
3.	MWNT/EPPGE	DPV	7.2	0.0009	5–100	Human blood plasma and urine	[105]
4.	CTAB-SnO2/GCE	CV	5	0.006 0.010	100–1000 1000–300,000	Human urine	[102]
5.	graphene-modified glassy carbon electrodes (GME)	CV	7	4000	600–120,000	Injection	[106]
6.	MoO3 NWs/GCE	CV	7	0.11	-	-	[101]
7.	GC/MWCNT/FCo98	CV	7	760	160–1910	~Drug	[100]
8.	bare BDD electrode	SWV CV	2	254	4900–490,000	Drug	[107]

**Table 11 materials-15-05782-t011:** Comparison of norepinephrine detection using CV and DPV based on bare boron doped electrodes with modifications.

S. No.	Analyte	Electrode	Range (nM)	LOD (nM)	Sample	Reference
1.	NE	ME/Au SAMs	2000–100,000	700	Drug	[108]
2.	NE	p-ATD/GCE	40–250	1700	Drug	[109]
3.	NE, AA, UA	PAAMWCNTs/SPCE	1000–10,000	130	-	[110]
4.	NE	CACE/GCE	550–9700	280	Drug	[111]
5.	NE	p-TMP/GCE	5000–100,000	80	Drug	[112]
6.	NE, DA	PL-Asp/GCE	30–16,000	4310	Drug	[113]
7.	NE, AC, FA	5ADMBCNPEs	15,000–1,000,000	8000	-	[114]
8.	NE	BDD	4900–490,000	1200	Drug	[107]

**Table 12 materials-15-05782-t012:** Comparison of norepinephrine detection using CV and DPV based on various electrodes with modifications.

S. No.	Electrode	Technique Used	Modifier	LDR (nM)	LOD (nM)	pH	Sample	Reference
1.	MIP	CV	Molecularly imprinted polymer-coated PdNPs	500–80,000	100	-	Real samples	[117]
2.	Glassy carbon	SWSV	Graphene quantum dots/gold nanoparticles	500–7500	150	7	Rat brain tissue	[116]
3.	Carbon paste	DPV	Graphene quantum dots/ionic liquid	20–400,000	60	7	Urine	[115]

## Data Availability

Not applicable.

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
