# Peer review of "Studies of Monoamine Neurotransmitters at Nanomolar Levels Using Carbon Material Electrodes: A Review"

_materials, 2022, doi:10.3390/ma15165782_

Round 1
Reviewer 1 Report
As we all know, hundreds of thousands of neurons exists in our human brain, and the neurotransmitters play an important role in transmission of the information between them. Accurately studying on the neurotransmitters is of great significance for understanding the function of the nervous system. In this manuscript, the authors summarized the recent advancements in electrochemical sensors for the detection of monoamine NTs in real samples at a nanomolar scale. It sorted out the ideas for our research in this field. The work is meaningful. It will provide a reference for subsequent research in NTs around the world. It can be accepted after minor revision.
1. The author said that carbon materrials have disadvantages like higher ET resistance. It seemed that surface modification can solve the problem. Are there any research about it?
2. Do the carbon material electrodes detect the only NTs for each electrode? Or whether we can detect three or more NTs by using one kind of electrode?
Author Response
As we all know, hundreds of thousands of neurons exists in our human brain, and the neurotransmitters play an important role in transmission of the information between them. Accurately studying on the neurotransmitters is of great significance for understanding the function of the nervous system. In this manuscript, the authors summarized the recent advancements in electrochemical sensors for the detection of monoamine NTs in real samples at a nanomolar scale. It sorted out the ideas for our research in this field. The work is meaningful. It will provide a reference for subsequent research in NTs around the world. It can be accepted after minor revision.
Dear reviewer, we are highly thankful for your careful analysis of our article. Based upon your suggestions we have improved our article. All the changes in the current modified version of the article are marked in red.
- The author said that carbon materials have disadvantages like higher ET resistance. It seemed that surface modification can solve the problem. Are there any research about it?
Dear reviewer, many reviews and research articles related to surface modification of electrodes for neurotransmitter detection are available (DOI: 10.1016/j.bios.2018.02.013, 10.1007/s00604-019-3315-y, 10.20964/2020.01.61)
- Do the carbon material electrodes detect only NTs for each electrode? Or whether we can detect three or more NTs by using one kind of electrode?
Dear reviewer as per your query till now simultaneous detection of 8 neurotransmitters has been done using high-performance liquid chromatography with fluorescence detection. (DOI: 10.3724/SP.J.1123.2011.00146). but carbon-based electrodes can simultaneously detect two neurotransmitters at a time (DOI: 10.1149/MA2022-01532195mtgabs), but detection of more than three simultaneously can be a great future direction using carbon materials.
Reviewer 2 Report
A reasonably well-written manuscript that covers carbon-based electrochemical sensors applied to the detection of monoamine neurotransmitters. This appears to be a niche updated version of a relatively recent review by Banerjee et al. [DOI:10.3390/bios10080101]. The review is relevant to a field of wide and important interest. It covers general aspects of NTs and their functions which are useful to an electrochemistry audience, at the same time the introduction of electrochemical methods is useful to readers primarily working in the neuroscience fields.
The following is a list of issues to address for revision: There is an apparent issue with the referencing of published papers. Figure 3 is taken in its entirety, including caption word-by-word, from ref. 15 without citation in the caption. Instead, it is referred to as adapted from ref. 14, as copied from ref. 15. In copying and pasting, the authors have also missed to realise that the scale unit should be micrometre not micromolar; and since arrows are shown, an explanation should be given in the caption for what they represent. It is very important to verify that there are no other missed or improper citations.
Lines 14-15: "Numerous research using" should be changed to "Numerous research studies using" since "research is singular.
Lines 17-18: "An investigation of the monoamine neurotransmitters is the primary goal of this review article." This statement is too general since the specific aim of the investigation is not specified, either remove or make it relevant to the goal of the review.
Line 27: It would read better after changing "no." to "number". Same for line 114.
Line 36: "Ach. More" what does it mean? Table 1: It would be useful to move this table to page 2 to help the reader refer to it. Also, define ELISA. Line 56: Please define L-DOPA.
Line 63: ST is introduced twice in this sentence, please remove one of the two.
Line 68: Please define SIDS.
Line 75: Please define LOD.
Line 118: Please define HPLC and GC-MS.
Figure 3B: In adapting Fig. 3 from ref. 14, the scale unit should be micrometre not micromolar, please correct this. Also, since you have left the arrows, please explain in the caption what they represent.
Line 204: "deep study" what is this meant to say about the study?
Line 250: "CPE" is never defined in the manuscript, please amend. In this regard, should " activated carbon modified carbon electrode (AC–CPE)" be " activated carbon modified carbon paste electrode (AC–CPE)"? NOTE: In general, please verify that all acronyms are fully defined in the manuscript, the reviewer stops here listing this issue further.
Line 336: Please correct this chemical reaction, there is a carbon atom in the products, but none in the reactants.
Line 876: This reaction sequence is not balanced, please check Contractions should be avoided in academic writing, please remove them, e.g. "it's" should the "it is", etc.
Author Response
A reasonably well-written manuscript that covers carbon-based electrochemical sensors applied to the detection of monoamine neurotransmitters. This appears to be a niche updated version of a relatively recent review by Banerjee et al. [DOI:10.3390/bios10080101]. The review is relevant to a field of wide and important interest. It covers general aspects of NTs and their functions which are useful to an electrochemistry audience, at the same time the introduction of electrochemical methods is useful to readers primarily working in the neuroscience fields.
The following is a list of issues to address for revision: There is an apparent issue with the referencing of published papers. Figure 3 is taken in its entirety, including caption word-by-word, from ref. 15 without citation in the caption. Instead, it is referred to as adapted from ref. 14, as copied from ref. 15. In copying and pasting, the authors have also missed to realise that the scale unit should be micrometre not micromolar; and since arrows are shown, an explanation should be given in the caption for what they represent. It is very important to verify that there are no other missed or improper citations.
Dear reviewer, we are highly thankful for your careful analysis of our article. Based upon your suggestions we have improved our article. All the changes in the current modified version of the article are marked in red.
Lines 14-15: "Numerous research using" should be changed to "Numerous research studies using" since "research is singular.
Based upon your suggestion we have changed “Numerous using to” "Numerous research studies using” in the lines 14-15 page 1.
Lines 17-18: "An investigation of the monoamine neurotransmitters is the primary goal of this review article." This statement is too general since the specific aim of the investigation is not specified, either remove or make it relevant to the goal of the review.
Based on your suggestion we have improved this statement as an investigation of the monoamine neurotransmitters at nanoscale using electrochemical methods is a primary goal of this review article. (line 17-18, page 1)
Line 27: It would read better after changing "no." to "number". Same for line 114.
Based upon your suggestion we have changed no. to the number on line 28, page 1, and line 118, page 2
Line 36: "Ach. More" what does it mean? Table 1: It would be useful to move this table to page 2 to help the reader refer to it. Also, define ELISA. Line 56: Please define L-DOPA.
Based on your suggestion we have defined Ach at line 37, page 1, ELISA at the table 1 caption (line 51, page 2), and L-DOPA at line 61, page 2.
Line 63: ST is introduced twice in this sentence, please remove one of the two.
Based upon your suggestion we have removed One of the ST at line 67, page 2.
Line 68: Please define SIDS.
Based on your suggestion we have defined SIDS at line 72, page 2.
Line 75: Please define LOD.
Based upon your suggestion we have expanded LOD in line 79 at page 2.
Line 118: Please define HPLC and GC-MS.
Based on your suggestion we have expanded the HPLC at the table 1 caption (line 52, page 2) and GC-MS in line 122 at page 3.
Figure 3B: In adapting Fig. 3 from ref. 14, the scale unit should be micrometre not micromolar, please correct this. Also, since you have left the arrows, please explain in the caption what they represent.
Based upon your suggestion we have removed the error and placed the correct ref 15 in the figure. And removed Fig 3 B. (page 6)
Line 204: "deep study" what is this meant to say about the study?
Based on your suggestion we have analyzed the sentence and removed the deep word in line 208.
Line 250: "CPE" is never defined in the manuscript, please amend. In this regard, should " activated carbon modified carbon electrode (AC–CPE)" be " activated carbon modified carbon paste electrode (AC–CPE)"? NOTE: In general, please verify that all acronyms are fully defined in the manuscript, the reviewer stops here listing this issue further.
Based on your suggestion we have defined CPE on line 254 and changed "activated carbon modified carbon electrode (AC–CPE)" to " activated carbon modified carbon paste electrode (AC–CPE) (line 262 ) on page 8.
Line 336: Please correct this chemical reaction, there is a carbon atom in the products, but none in the reactants.
Based upon your suggestion we have corrected the error in the equation and have removed C from the equation. (line 339, page number 10)
Line 876: This reaction sequence is not balanced, please check Contractions should be avoided in academic writing, please remove them, e.g. "it's" should the "it is", etc.
Based upon your suggestion we have removed the contraction on page 34 (line number 889) and replaced it.